# Integrated histopathology of the human pancreas throughout stages of type 1 diabetes progression

Verena van der Heide [1,10], Sara McArdle [2,10], Michael S. Nelson [3], Karen Cerosaletti [4], Sacha Gnjatic [1,5], Zbigniew Mikulski[2], Amanda L. Posgai [6], Irina Kusmartseva [6], Mark A. Atkinson [6,7] & Dirk Homann [1,8,9] ✉

Type 1 diabetes (T1D) is a progressive autoimmune condition that culminates in loss of insulin-producing beta cells. Pancreatic histopathology provides essential insights into disease initiation/progression yet an integrated perspective onto in situ pathogenic processes is lacking. Here, we combined multiplexed immunostaining, high-magnification whole-slide imaging, digital pathology, and semi-automated image analyses to interrogate pancreatic tail and head sections across T1D stages, including at-risk and at-onset cases. Deconvolution of architectural features, endocrine cell composition, immune cell burden, and spatial relations of ~25,000 islets effectively contextualizes previously established and additional pancreatic hallmarks in health and T1D. Our results reveal a spatially homogenous and islet size-contingent architectural organization of the endocrine pancreas, a notable coordination of organ-wide pathogenic processes, and multiple histopathological correlates that foreshadow distinctive T1D histopathology already at the preclinical stage. Altogether, we propose a revised natural history of T1D with implications for further histopathological investigations and considerations of pathogenetic modalities.

Type 1 diabetes (T1D) is a chronic autoimmune disease characterized by the destruction of insulin-producing beta cells in pancreatic islets[1,2]. An interplay of genetic susceptibility[3], environmental factors[4], and immune dysregulation[5] promotes multiple pathological alterations including pancreatic inflammation, islet-associated immune cell infiltration (insulitis), exocrine abnormalities, and systemic complications due to overt hyperglycemia[6–8]. Clinically, T1D progresses through three major stages: emergence of ≥2 T1D-associated islet auto-antibodies but maintenance of normoglycemia (stage 1); additional development of dysglycemia (stage 2); and eventual clinical disease (stage 3)[9].

Early histological interrogations of the pancreas established T1D as an autoimmune disease[10] and defined pathological hallmarks[11], yet extensive in situ studies have mostly been limited to the past ~15 years

---

[1]Marc and Jennifer Lipschultz Precision Immunology Institute, Department of Immunology and Immunotherapy, Icahn School of Medicine at Mount Sinai (ISMMS), New York, NY, USA. [2]Microscopy and Histology Core Facility, La Jolla Institute for Immunology, La Jolla, CA, USA. [3]Department of Biomedical Engineering, University of Wisconsin-Madison, Madison, WI, USA. [4]Center for Translational Immunology, Benaroya Research Institute, Seattle, WA, USA. [5]Tisch Cancer Institute, Department of Medicine, ISMMS, New York, NY, USA. [6]Department of Pathology, Immunology, and Laboratory Medicine, University of Florida Diabetes Institute, College of Medicine, Gainesville, FL, USA. [7]Department of Pediatrics, University of Florida Diabetes Institute, College of Medicine, Gainesville, FL, USA. [8]Diabetes, Obesity & Metabolism Institute, Department of Medicine, ISMMS, New York, NY, USA. [9]Diabetes Research Institute, University of Miami, Miami, FL, USA. [10]These authors contributed equally: Verena van der Heide, Sara McArdle. ✉e-mail: dxh2185@miami.edu

following the establishment of specialized tissue repositories such as the Network for Pancreatic Organ donors with Diabetes (nPOD) program[12]. Nevertheless, estimates suggest that <700 T1D donor pancreata are available for research globally[13], a limited resource that is further compounded by substantial heterogeneity at the level of organ donors, pancreas anatomy (*e.g.*, organ region, lobular insulitis), islet properties (*e.g.*, architecture, endocrine composition, association with immune cells), and altered endocrine function beyond beta cells (*e.g.*, alpha cells[14,15]). Pancreata from donors at high risk (stage 1/2, elevated HLA class-II risk) and at-onset of symptomatic T1D, critical for pathogenesis studies, are exceptionally scarce.

In order to contend with these challenges, emerging investigative techniques such as whole-slide imaging[11,16–27], high-dimensional tissue analysis[26,28–34], and three-dimensional (3D) visualization of immuno-stained thick pancreas slices[35–37] have been deployed in conjunction with tailored digital analysis software. However, to date no standardized data analytics platform dedicated to pancreatic histopathology is readily accessible. Commercial platforms featuring "pancreas modules" and machine learning algorithms have aided the identification of pancreas and islet features[19,21,26,31,38–40], but their proprietary nature restricts customization.

To our knowledge, a broad integration of the complementary strands of histopathological T1D pancreas investigation and digital pathology modalities[12,41] has not yet been performed. To this extent, we combined higher dimensional multiplex brightfield immunohistochemistry (IHC), whole-slide image acquisition at high magnification, and a semi-automated analysis pipeline implemented in QuPath, a versatile open-source digital pathology platform[42,43] that has been employed in recent T1D studies primarily focused on immune cell analyses[22–25,44]. Our results reveal a spatially homogenous and islet size-contingent architectural organization of the endocrine pancreas; they demonstrate a notable coordination of tissue-wide pathogenic processes characterized by alterations of islet properties independent of associated immune cell burden; and they document the presence of distinct histopathological disease correlates already at preclinical stage 1/2 T1D. Accordingly, we here propose a revised natural history of T1D as a chronic-progressive condition that readily contextualizes established hallmarks and additional observations of T1D pathology.

## Results

### Multiplexed IHC analysis of human pancreatic tissue sections
The detailed histopathological interrogation of healthy and diabetic human pancreatic sections remains a cornerstone of investigations into T1D pathogenesis[12]. To leverage recent advances in tissue imaging, we employed the Multiplexed Immunohistochemical Consecutive Staining on Single Slide (MICSSS) platform[45,46], an iterative IHC assay that permits successive brightfield visualization of immunostaining patterns at high magnification (40x) across whole slides (Fig. 1a). Previous MICSSS applications were largely restricted to characterization of CD45[+] hematopoietic cells[47–49], and our adaptation for the visualization of eight pancreatic endocrine hormones in addition to CD45, in part due to larger intracellular staining areas, required adjustments to signal amplification, destaining/blocking, and a deliberate determination of staining order since our targets of interest exhibited differential sensitivity to deterioration over repeated tissue processing steps (Figs. 1b and S1a).

### Development of a semi-automated image analysis pipeline
We used machine learning to analyze pancreatic whole-slide images and customized a series of specific tasks in the open-source digital pathology software QuPath[42]. We first captured total tissue specimen areas and excluded surrounding connective tissue and fat to demarcate parenchymal (exocrine and endocrine) areas (Fig. 1c). Alignment of nine immuno-stained images per tissue section was performed using affine transformation and achieved cell-level accuracy.

Pancreatic islets, defined here as endocrine cell clusters $\geq 1000\ \mu m^2$ (~36 μm diameter), were delineated by merging the areas of six selected hormone stains with residual non-endocrine areas to create contiguous objects (Fig. 1d and Methods). Geometric and spatial properties of islet outlines were quantified to allow for a basic description of islet architecture and derivation of shape descriptors (Figs. 1e and S1b) some of which reportedly change[28,50,51] or not[52] with T1D progression.

To delimit hormone staining areas within each islet, islet outlines were superimposed onto each image and a pixel classifier was used to segment positive staining areas. Given the complete lack of background signal and heterogeneous pattern of proinsulin (ProINS) expression as observed previously (high: Golgi/immature INS granules; low: cytoplasm/ER)[53,54], we quantified islet areas of both high and total ProINS content (Fig. 1f). Representative images for other hormone area captures are shown in Fig. S1c. However, due to creeping introduction of some background signal over successive staining rounds, we refrained from distinguishing areas of low/high expression for all other hormones thus permitting proper demarcation of mutually exclusive staining areas (Fig. 1g). The segmented regions from each hormone stain were further used to calculate total endocrine areas (*i.e.*, union of eight hormone stains) and Jaccard indices, a statistic that quantifies the overlap of brightfield staining areas and corroborates the accuracy of our image alignment/segmentation strategy (Fig. S1d). Lastly, we incorporated the StarDist algorithm[55] and an object classifier into our analysis pipeline to capture CD45[+] immune cells within islets and the surrounding peri-islet areas (Figs. 1g and S1c).

The importance of accounting for pancreatic polypeptide (PPY) expression, largely segregated to the uncinate process of the pancreatic head (PH)[56–58], is illustrated by our observation that some PPY[+] islets lack alpha and beta cells even in non-diabetic donors (Figs. 1h and S1e); thus, any consideration of insulin-deficient islets (IDIs) in the PH needs to include PPY analyses. Altogether, automation of multiple image analysis tasks in combination with multiplexed whole-slide image acquisition offers unique opportunities to revisit pancreatic histology in health and T1D disease *at scale*.

### Pancreas donors and cumulative endocrine contents across T1D stages
To interrogate the histopathological evolution of T1D, we performed MICSSS staining of pancreatic tail (PT) and PH sections from four donor groups: non-diabetic controls (Ctrl), autoantibody-positive stage 1/2 T1D (AAb), short duration stage 3 T1D (T1DS; <2 years including three at-onset donors), and longer duration T1D (T1DL; 8–11 years) (Table 1 and Supplementary Data 1). Donor matching was performed on age and gender and, where possible, demographic (ethnicity) and clinical (body mass index/BMI) parameters; several outliers in our cohorts are described in Supplementary Data 1. HLA-II risk[59–62] was calculated to be low for Ctrl donors and similarly elevated for AAb, T1DS and T1DL donors (Supplementary Data 1 and Fig. S2a). Donor age, BMI, and total pancreas weight as well as other clinical parameters displayed significant positive correlations as expected[63] (Fig. S2b, c).

Consistent with the T1D-associated loss of pancreas mass[63,64], relative pancreas weights (RPW) tend to decline in T1DS donors, with a more pronounced reduction in our T1DL cohort (Fig. S2d), a trend also reflected in a slight decrease of parenchymal tissue section areas; PT but not PH parenchymal areas further correlate positively with age (Fig. S2e, f). On average, we captured ~500 islets per tissue section, with absolute numbers broadly decreasing as T1D progresses (Fig. S2g). While islet densities show a similar 1.5–2.0-fold non-significant reduction across donor cohorts, cumulative islet areas decline significantly (Ctrl *vs.* T1DL PH: 1.8-fold; PT: 2.9-fold) corresponding to an overall 2.9–5.1-fold loss of islet mass (Figs. 1i and S2h; *cf.* Supplementary Data 2). Quantifying cumulative islet hormone staining areas as a fraction of parenchymal tissue areas provides an initial

orientation about disease-associated alterations of pancreatic endocrine content. As in earlier whole-slide imaging studies[16,17], insulin (INS) staining areas are significantly reduced in T1DS/T1DL donors while glucagon (GCG) areas remain largely preserved (Fig. S2i). Several additional observations are noteworthy: a ~ 2.7-fold loss of chromogranin A (CHGA) from Ctrl to T1DL stage accompanies the decline of relative islet areas; total ProINS areas are larger than INS areas whereas ProINS[hi] areas are smaller and comparable to a previous report[17]; a

decrease of ProINS and islet amyloid polypeptide (IAPP) areas in AAb donors (significant for PH but not PT) strikingly echoes recent findings about lower regional beta cell volume assessed by 3D morphometric analyses[36]; somatostatin (SST) staining areas remain unaltered in all donor groups; and PPY expression, minor in PT but prominent in PH, also appears unaffected by disease progression (Fig. S2i).

For additional context, we document a progressive decline of INS:GCG expression ratios with disease advancement (Fig. S2j) as well

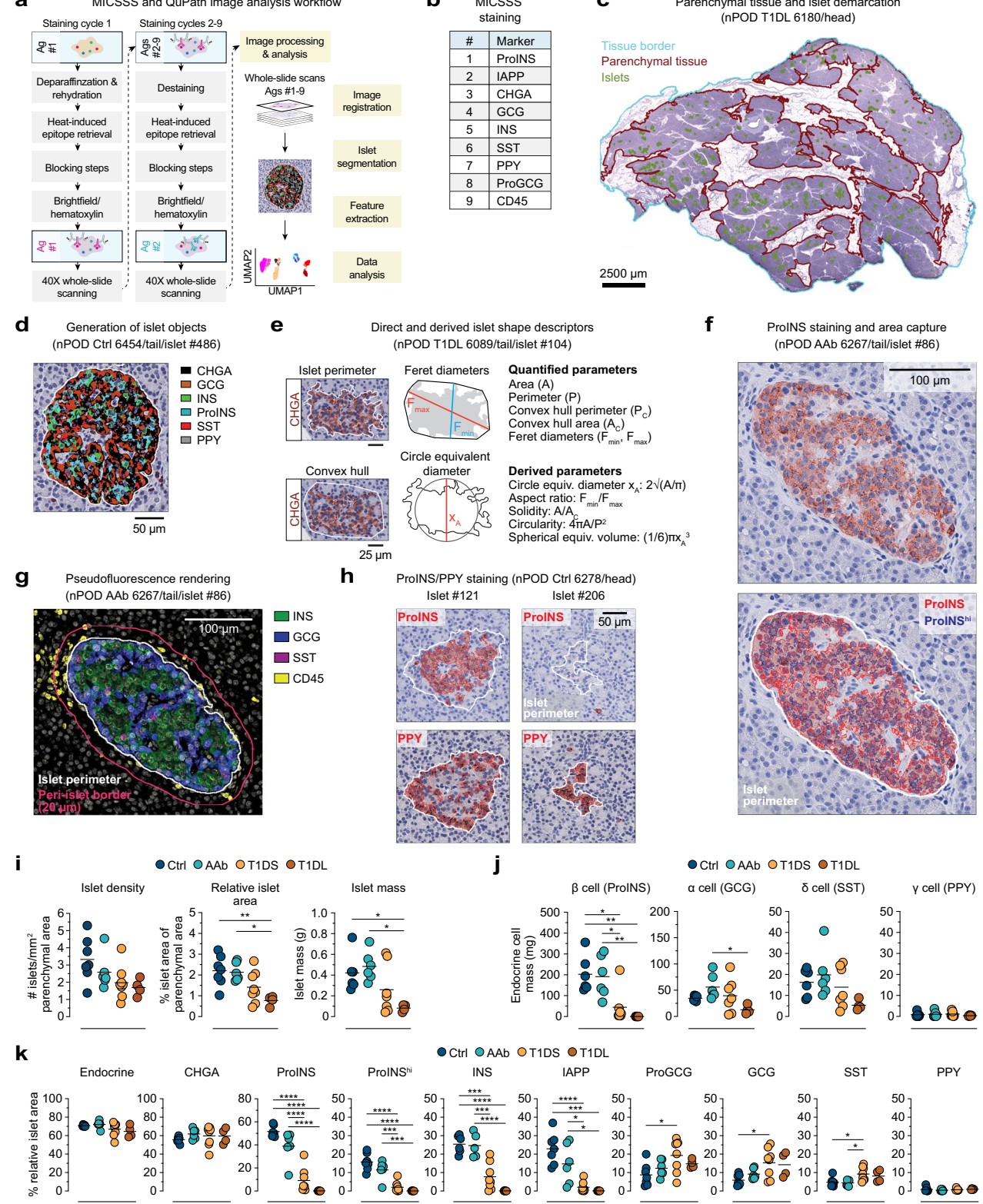

**a** MICSSS and QuPath image analysis workflow

**b** MICSSS staining

| # | Marker |
|---|--------|
| 1 | ProINS |
| 2 | IAPP |
| 3 | CHGA |
| 4 | GCG |
| 5 | INS |
| 6 | SST |
| 7 | PPY |
| 8 | ProGCG |
| 9 | CD45 |

**c** Parenchymal tissue and islet demarcation (nPOD T1DL 6180/head)

**d** Generation of islet objects (nPOD Ctrl 6454/tail/islet #486)

**e** Direct and derived islet shape descriptors (nPOD T1DL 6089/tail/islet #104)

**f** ProINS staining and area capture (nPOD AAb 6267/tail/islet #86)

**g** Pseudofluorescence rendering (nPOD AAb 6267/tail/islet #86)

**h** ProINS/PPY staining (nPOD Ctrl 6278/head)

**i** Ctrl AAb T1DS T1DL — Islet density; Relative islet area; Islet mass

**j** Ctrl AAb T1DS T1DL — β cell (ProINS); α cell (GCG); δ cell (SST); γ cell (PPY); Endocrine cell mass (mg)

**k** Ctrl AAb T1DS T1DL — Endocrine, CHGA, ProINS, ProINS[hi], INS, IAPP, ProGCG, GCG, SST, PPY; % relative islet area

**Fig. 1 | MICSSS staining, semi-automated image analysis strategies, and basic properties of pancreatic tissue sections and islets. a** Workflow of MICSSS staining and subsequent image analysis pipeline. **b** MICSSS staining order. In subsequent panels c–h, donor type (Ctrl: non-diabetic, AAb: stage 1/2 T1D, T1DL: long-duration stage 3 T1D), nPOD ID and pancreatic region are indicated. **c** Demarcation of total tissue, parenchymal (exocrine/endocrine) tissue and islet areas. **d** Generation of islet objects by additive display of six endocrine hormone stains (no PPY in present islet). **e** Islet shape measurements and equations for derived parameters. **f** ProINS brightfield stain (brown), automated capture of ProINS staining areas (ProINS$^{total}$: red traces, ProINS$^{hi}$: blue traces), and islet perimeter (white). **g** Pseudo-fluorescent rendering of overlaid INS, GCG, SST and CD45 stains, and demarcation of peri-islet region (border at 20 μm distance from islet perimeter). **h** ProINS and PPY staining (brown) and respective area captures

(red traces) for representative ProINS$^+$PPY$^+$ and ProINS$^-$PPY$^+$ islets. **i** Islet densities, relative areas and mass in PT. **j** Endocrine cell type mass in PT was calculated from data in Fig. S2i and pancreas region weights (Supplementary Data 1) excluding the two 5-year-old donors (Ctrl 6382, T1DS 6209). **k** Fraction of indicated endocrine hormone staining areas in PT islets ("endocrine": union of all hormone staining areas). Due to notably weak CHGA (Ctrl 6162) and/or INS (Ctrl 6162, AAb 6450) staining of PT sections, respective data are excluded. Colored circles in scatter plots in (**i**–**k**) display mean values derived from individual donors and black horizontal bars are donor group means with n = 7 Ctrl, 6 AAb, 8 T1DS and 4 T1DL donors (*cf.*, Table 1). Statistical analyses in (**i**–**k**) were conducted with ordinary one-way ANOVA and Tukey's multiple comparisons test (*$p < 0.05$, **$p < 0.01$, ***$p < 0.001$, ****$p < 0.0001$).

## Table 1 | Overview of pancreatic organ donors

| nPOD donors<br>Number of subjects | Ctrl<br>7 | AAb<br>6 | T1DS < 2 yrs<br>8 | T1DL 8–11 yrs<br>4 |
|---|---|---|---|---|
| Age in years (mean ± SD) | 17.5 (± 7.6) | 22.5 (± 3.3) | 17.3 (± 8.3) | 19.6 (± 7.5) |
| Age (years) at T1D onset (mean ± SD) | n/a | n/a | 16.4 (± 8.2) | 9.8 (± 6.2) |
| Female/Male (%) | 3/4 (43/57) | 3/3 (50/50) | 5/3 (63/38) | 1/3 (25/75) |
| Body Mass Index (mean ± SD) | 24.3 (± 4.9) | 28.3 (± 11.7) | 22.8 (± 9.3) | 25.1 (± 2.1) |
| Autoantibody number (mean ± SD) | 0 | 1.8 (± 0.4) | 1.9 (± 1.6) | 2.3 (± 2.1) |
| T1D duration in years (mean ± SD) | n/a | n/a | 0.6 (± 0.9) | 9.8 (± 1.5) |

**Ctrl:** non-diabetic control donors (outlier: Ctrl 6382 - young child ~5 years of age).
**AAb:** stage 1/2 T1D donors with ≥ 2 autoantibodies (no distinction of stage 1 and 2 T1D in the absence of metabolic tests; outliers: AAb 6424 – severe obesity, and AAb 6310 - only 1 AAb but included here due to previously determined presence of insulitis).
**T1DS:** short duration stage 3 T1D (outliers: T1DS 6209 - young child ~5 years of age, T1DS 6405 – severe obesity).
**T1DL:** long duration stage 3 T1D.
For further pancreatic specimen and organ donor details, see Supplementary Data 1.

as an estimation of total endocrine cell type mass. Beta cell mass wanes precipitously (Ctrl *vs.* T1DL ProINS: ~780-fold; INS: ~1250-fold), but both alpha- (ProGCG/GCG) and delta- (SST) cell mass also decrease in T1DL donors, albeit to a lesser degree ( ~ 3-fold; PT > PH). In contrast, gamma cell (PPY) mass is minimal in the PT ( ~ 1 mg) yet substantial and variable in the PH ( ~ 20–40 mg), presumably due to different uncinate process proportions[56,57] in our tissue sections (Figs. 1j and S2K). These findings readily recapitulate and refine the histopathological dynamics of T1D progression (including loss of ~70–80% beta cell mass in T1DS[65]); they emphasize a considerable alteration of the endocrine pancreas in the T1DL stage beyond loss of beta cells; and they provide a general framework for subsequent analyses focused on ~25,000 individual islets captured across the four donor cohorts.

### Islet endocrine hormone contents throughout T1D progression

Extending our quantification of pancreatic endocrine contents to individual islets reveals commonalities and differences compared to cumulative staining area assessments. A profound reduction of ProINS, INS and IAPP contents in T1DS islets is similarly presaged by a trend towards decreased ProINS and IAPP but not INS expression in AAb donors; T1DS and T1DL islets present with a relative compensatory increase of ProGCG, GCG and SST expression areas due to loss of beta cells; and the variable PPY$^+$ proportion of PH islets appears particularly large in T1DL donors (Fig. 1k and S2l). At the same time, relative endocrine and CHGA areas remain constant throughout T1D progression thus corroborating, in conjunction with the ~2.7-fold reduction of total CHGA staining area (Fig. S2i), a net islet loss in T1DL. A direct comparison of PT and PH islets demonstrates that in all donor groups, despite comparable islet endocrine areas, PH islets express less CHGA, ProINS, INS, ProGCG, and GCG but equivalent IAPP and SST (Fig. S3a). While reduced alpha and beta cell mass in the PH of Ctrl donors has been noted earlier[57,66], quantifying average islet hormone contents cannot resolve the potential contribution of different islet

subsets to what appears an overall distinctive PH endocrine composition.

### Frequencies of endocrine content-stratified islet subsets

To evaluate how islet heterogeneity shapes net endocrine cell mass, we quantified islet frequencies based on their hormone content or lack thereof (<1% hormone staining area of islet area[35]). The significant increase of IDIs (ProINS$^{neg}$ or INS$^{neg}$) with disease development (T1DS PT: ~64%, PH: 84%; T1DL PT/PH: >99%) constitutes the histopathological hallmark of T1D (Figs. 2a and S3b). An unexpected feature of the non-diabetic pancreas revealed by recent 3D mapping is the abundance of small "GCG-deficient" islets[35]. We confirm this observation in our 2D interrogation by demonstrating that ~34% (PT) to ~56% (PH) of Ctrl islets are "GCG-deficient", and that ProINS$^+$GCG$^{neg}$ islets are significantly smaller than ProINS$^+$GCG$^+$ islets (Figs. 2b, c and S3c, d). The absence of alpha cells, as recently shown with human pseudo-islets exclusively composed of beta cells, does not appear to impinge on their functionality[67], corroborating the notion that "GCG-deficient" islets are an integral part of endocrine pancreas physiology. Importantly, the ProINS$^+$GCG$^{neg}$ islet fraction declines with T1D progression, including a significant reduction in AAb *vs.* Ctrl donor PTs, indicating that this subset is particularly vulnerable in early T1D pathogenesis (Figs. 2c and S3d).

Furthermore, a differential abundance of PPY$^+$ islets in PT *vs.* PH aligns with cumulative PPY staining patterns (Fig. 2d), and a subset stratification according to ProINS/PPY expression demonstrates that the Ctrl PH contains a ~10% fraction of small PPY$^+$ islets lacking beta cells that increases to ~60% in T1DL (Fig. 2e). Delta cells are found in ~65% of islets, and their relative abundance is somewhat elevated in clinical T1D as noted before[68] (Figs. 2f and S3e). Lastly, we traced hormone co-expression patterns in islets across disease course. For alpha cell hormones, a modest increase of Jaccard indices for CHGA, ProGCG and GCG with T1D progression is consistent with the relative

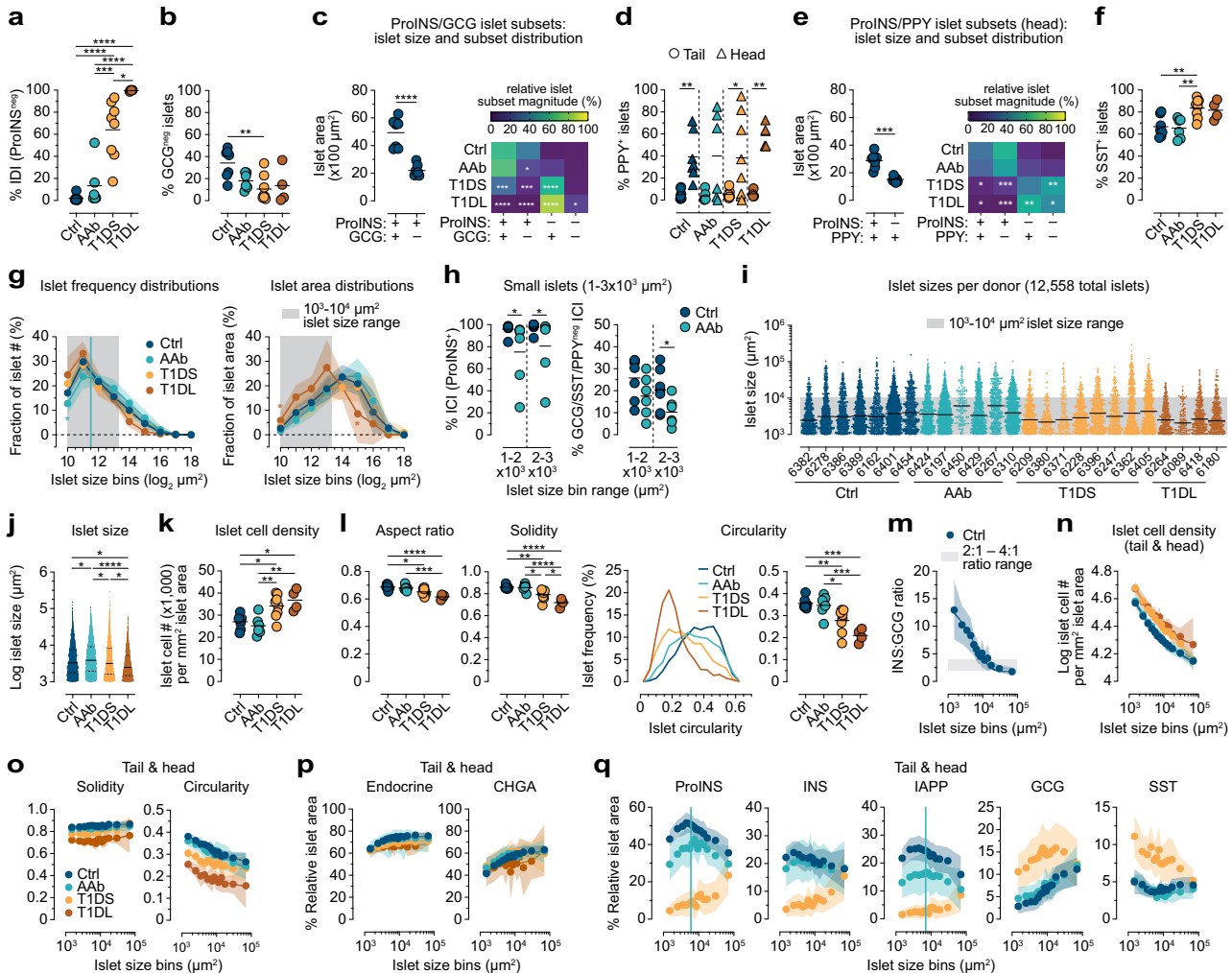

**Fig. 2 | Islet and islet subset properties across stages of T1D progression.** Unless noted otherwise, data is derived from PT sections displaying scatter (individual donor means) and group means (black bars) with n = 7 Ctrl, 6 AAb, 8 T1DS and 4 T1DL donors. **a** Frequencies of IDIs (< 1% ProINS content). **b** Frequencies of "GCG-deficient" islets (< 1% GCG content). **c** Left: median islet subset sizes in individual Ctrl donors and group means; right: heatmap displaying relative magnitude of indicated ProINS/GCG islet subsets in each T1D stage (asterisks pertain to comparisons of Ctrl *vs.* other donors). **d** Frequencies of islets with ≥1% PPY content. **e** Data display for combinatorial ProINS/PPY expression by PH islets as in (**c**). **f** Frequencies of islets with ≥1% SST content. **g** Islet frequencies (left) and respective contributions to overall islet area (right) as a function of log₂-transformed, bin-stratified islet sizes (mean±95% confidence interval [CI]; color-coded asterisks pertain to comparisons with Ctrl donors; vertical line set at 3000 µm²). **h** Fraction of Ctrl *vs.* AAb islet subsets in indicated islet size bins. **i** Size distribution of all 12,558 PT islets stratified according to donor group (individual donors ordered according

to increasing age; bars: medians). **j** Islet size distributions (group median/quartiles indicated). **k** Geometric means of islet cell densities. **l** Islet aspect ratio, solidity and circularity; binned histograms display islet circularity distributions. **m–q** Properties of combined PT/PH islets from individual donors in indicated groups are displayed as a function of islet sizes stratified into 14 bins (mean ± 95% CI; data exclusions: PT CHGA and/or INS data for Ctrl 6162/AAb 6450 [*cf.*, Fig.1k] and bins containing <3 islets). Lines in m-p represent exponential associations; goodness of curve fits and additional parameters shown in Fig. S4a–h. Statistical analyses conducted with two-tailed Student's *t* test (paired: **c**/left, **d**, **e**/left; unpaired: **h**/right), Mann-Whitney U test (**h**/left), one-way ANOVA with Tukey's multiple comparisons test (**a**, **b**, **c**/right, **e**/right, **f**, **g** [bin-specific lognormal ANOVA], **k**, **l**, **n–q**; in **q**, ProINS and IAPP content is significantly lower for AAb *vs.* Ctrl islets <6000–7000 µm² [vertical lines; *p* < 0.03]), or mixed model analyses (**j**, see Methods) (**p* < 0.05, ***p* < 0.01, ****p* < 0.001, *****p* < 0.0001).

rise of ProGCG/GCG content in islets afflicted by beta cell loss. For beta cell hormones, Jaccard indices slightly decline in AAb donors before markedly plunging with T1D onset; this decrease is more pronounced than that of individual hormone staining areas reflecting profound perturbations of INS synthesis[69] (Fig. S3f).

## Islet architecture and its histopathological alterations

The geometric properties of islets have mostly been studied in non-diabetic pancreata[70–73] and in accordance with these reports, Ctrl islet areas span across two orders of magnitude ($10^3$ – $10^5$ µm²) following a frequency distribution that is notably skewed toward smaller islets: ~82% of PT and PH islets are found in the $10^3$–$10^4$ µm² range; due to their smaller size, however, they respectively contribute only ~41–50%

to overall islet area (Figs. 2g and S3g). Although these distribution patterns are roughly maintained throughout T1D progression, disease stage-specific deviations are discernible especially in the PT where islet size positively correlates with age (Fig. S3h). A relative loss of the smallest islets (< 3000 µm²) in AAb *vs.* Ctrl donors (Fig. 2g) is notable since they account for ~50% of all islets in Ctrl donors and their corresponding density is decreased by ~1.6-fold in AAb donors. A subtle rightward extension and flattening of the T1DS islet area distribution curve hints at preferential preservation of larger islets, and the more pronounced left-shift of the corresponding T1DL curve indicates an overall islet size reduction (Fig. 2g). Furthermore, the remaining small AAb islets are partially compromised due to some beta cell-depletion (*i.e.*, reduction of the ProINS⁺ islet fraction) and a relative decrease of

"GCG/SST/PPY-deficient" insulin-containing islets (ICIs) (Fig. 2h) that are potentially marked for complete elimination following subsequent beta cell destruction at stage 3 T1D.

Consistent with an overall small islet loss in AAb donors, mixed model analyses demonstrate a significant increase of median islet size in stage 1/2 T1D (Fig. 2i, j). A subsequent decline of median islet size in T1DS and especially T1DL donors (Fig. 2j) is further accompanied by greater cellular densities in islets (Fig. 2k), confirming an earlier report[21] and conceivably reflecting local deprivation of trophic INS effects previously invoked as cause for exocrine pancreas shrinkage[64]. Therefore, a T1D-associated islet size decrease accompanies the reduction of overall islet densities (Fig. 1i; see also ref. 52) thus compounding the loss of islet mass. These patterns are not apparent in the PH (Fig. S3g–j; in part due to the preponderance of PPY⁺ IDIs, Fig. 2e), yet higher cellular densities in T1DS/T1DL islets are comparable for both PT/PH compartments (Figs. 2k and S3j).

Recent 3D mapping of INS- and GCG-staining volumes in non-diabetic donors demonstrated that average human islet size is smaller than previously appreciated[35]. For comparative purposes, and working with similar assumptions about circularity and sphericity of islet objects, we calculated median/mean PT/PH islet diameters and volumes from our islet area measurements yielding Ctrl group estimates that are in good alignment with islet size assessments in ref. 35 (Fig. S3k). Similarly, while islets with >91 µm diameter constituted ~26% of all islets yet contributed ~75% to beta cell mass in that 3D study[35], we find that islets above average size (>89 µm diameter) account for ~28% of islets (a proportion that is near identical across PT and PH as well as disease stages) and contribute ~70% to islet mass in all donor groups (Fig. S3l).

In contrast to the subtle alterations of islet areas and a trend towards increased islet perimeters despite declining islet sizes with advancing disease (Fig. S3m), the progression of two-dimensional shape descriptors is profound: islet aspect ratio, solidity, and circularity significantly decline from Ctrl and AAb to T1DS and T1DL stage, a pattern that again appears somewhat more accentuated for PT than PH islets (Figs. 2l and S3n; aspect ratio, solidity and circularity are dimensionless measurements in the range of 0–1 that respectively describe object elongation, overall concavity, and similarity to a circular object; equations in Fig. 1e). Together, these alterations signify a stark deterioration of islet architecture, likely a direct consequence of beta cell destruction. Our results indicate that inclusive whole-slide image analyses capture basic islet architectural and hormone expression features that readily align even with contemporary volumetric measurements[35–37] and thus imbue the scope of observations reported here with greater confidence.

## Basic exponential associations between islet size and other islet properties

The heterogeneity of islet composition has long been recognized[65,74] and in an attempt to account for disparate islet properties (including the conspicuous lack of alpha and/or beta cells in smaller Ctrl islets), we considered islet features as a function of their size[75,76] (Fig. S4a–h). Correlating islet size frequency distributions in Ctrl pancreata with specific islet properties, we find that simple exponential relations consistently capture these associations: increasing islet size correlates with greater islet cell numbers, solidity, endocrine/CHGA content and alpha cells but lower islet cell density, circularity as well as beta- and gamma cell contents (these patterns only diverge for the smallest islets with comparatively reduced beta cell fractions, and for SST content which is slightly elevated in both smaller and larger islets) (Fig. S4a, b). Consequently, islet INS:GCG expression ratios decline with increasing islet size (Fig. 2m). Overall, this approach provides a simple organizing principle for islet heterogeneity that may be particularly useful for islet comparisons across disease stages.

Throughout T1D progression, the exponential associations of islet size and architectural features are subject to a modest, largely size-independent upward shift (cellular density) or downward shift (solidity and circularity, somewhat sparing T1DS islets >10,000 µm²) of T1D stage-specific distribution curves indicative of pathogenetic processes encompassing islets of all sizes (Figs. 2n, o and S4a, c, e, g). At the same time, corresponding relations for endocrine and CHGA contents remain strikingly resilient over T1D course (Figs. 2p and S4b, d, f, h). While correlations of islet size with endocrine cell type-specific hormone abundance also remain broadly similar in Ctrl *vs.* AAb donors, a particular reduction of relative ProINS and IAPP but not INS proportions is discernible especially for AAb islets <6000 µm² (Figs. 2q and S4d). The subsequent T1DS stage features three deviations from these patterns: an overall reduced yet relatively higher ProINS, INS and IAPP expression by larger islets, proportionally more ProGCG/GCG expression for all but the largest islets, and a decline of SST expression as a function of islet size (Figs. 2q and S4f). In T1DL donors, these patterns, beyond the profound beta cell loss, are exacerbated as reflected by the considerable variability of islet size-associated ProGCG, GCG and PPY contents (Fig. S4h). The dynamic regulation of these exponential relations reflects an early targeting of beta cells especially in smaller islets, a temporary preservation of residual beta cell mass in larger T1DS islets, and the markedly altered islet composition in T1DL donors (Fig. S4i). Collectively, our semi-automated whole-slide image analyses confirm central tenets of T1D pathology and readily identify multiple additional islet properties in normal pancreata and over the entire course of T1D.

## Integrated histopathology throughout T1D progression

The true analytical potential of our approach to characterization of islet heterogeneity lies in elucidating complex combinatorial property patterns. We therefore adapted a dimensionality reduction tool commonly employed for transcriptomic single-cell analyses and performed UMAP clustering[77] of all ~25,000 individual islets (rather than single-cells) captured in our four donor cohorts based on islet architectural and hormone expression properties as well as CD45⁺ cell associations and neighborhood relations (see Methods). This strategy generates five major clusters (I–V), four of which contain 3–4 subclusters (A-D), and a regional stratification illustrates that clusters III and IV are predominantly found in the PH, underscoring apparent differences in the broad organization of PH and PT endocrine compartments (Fig. 3a).

Visualization of individual hormone expression levels across these clusters emphasizes both shared and distinct properties of cluster-constituent islets: endocrine areas are comparable for most clusters though slightly diminished for clusters III/IV, a pattern that is more pronounced for CHGA expression which is somewhat decreased in cluster III and strongly reduced in cluster IV (Fig. 3b, c). Beta-cell-containing islets are found exclusively in clusters I/II/III; alpha-cells populate the majority but not all cluster I/II islets, are absent from cluster III and very infrequent in cluster IV, but a prominent presence in clusters V-A/BC; delta-cells are distributed across all major clusters; and PPY⁺ islets, outside a smattering found in clusters I/II, reside in cluster III and dominate cluster IV (Fig. 3c). The islet feature histograms and heatmaps in Figs. 3d and S4j provide a more granular perspective on distinct cluster properties by resolving architectural and hormone expression properties at the level of individual donors; they are complemented by various scatter plot arrangements to illustrate more subtle alterations and allow for clarification of significant differences between UMAP clusters and disease stages (Fig. S5a–j, which serves as a consolidated data resource).

Cluster stratification shows a hierarchy of average islet sizes (cluster II >> I > V-A > III ~ IV > V-BC) with near identical islet dimensions for respective PT and PH clusters (Figs. 3d and S5a, c, i) indicating that

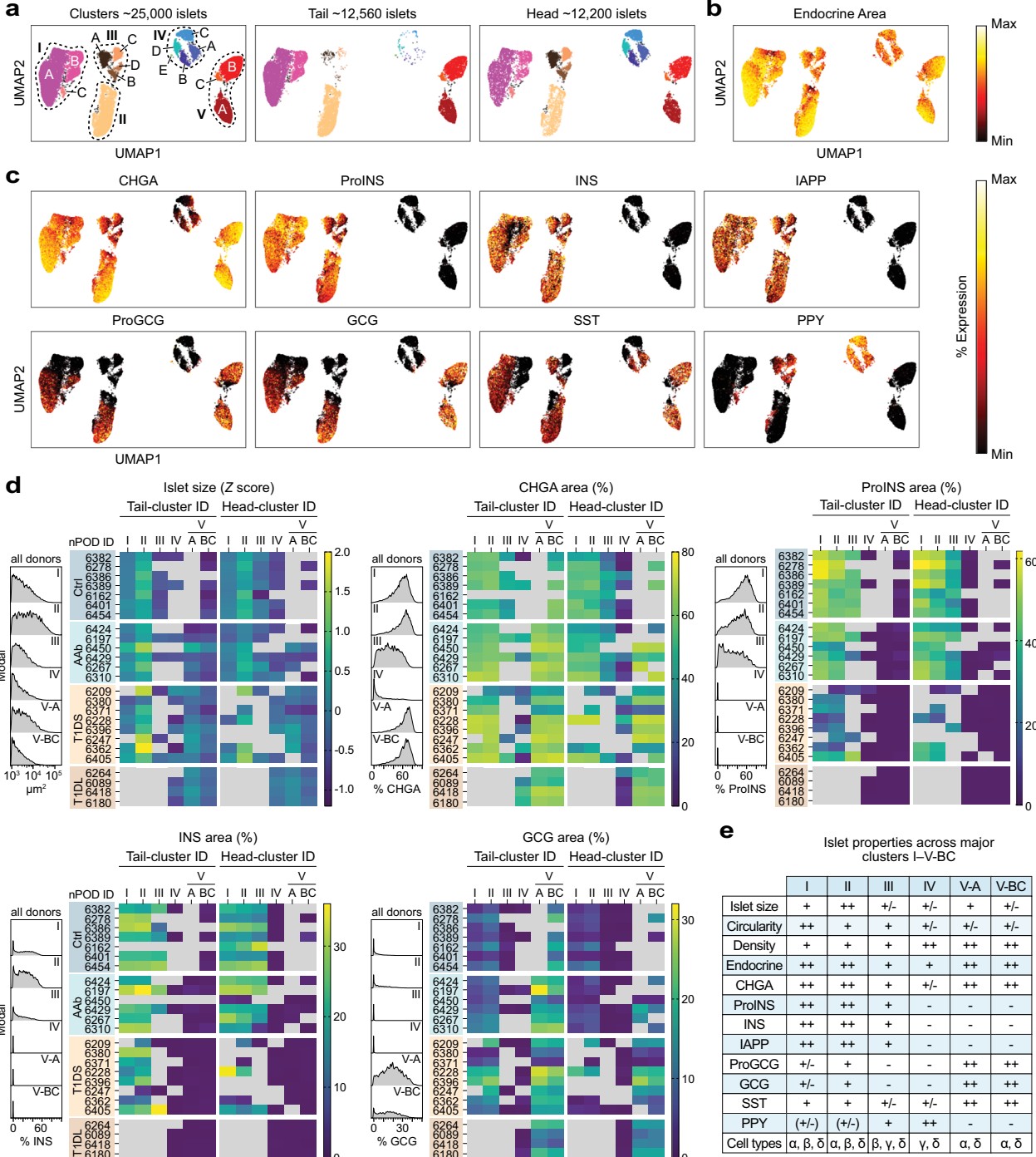

**Fig. 3 | UMAP clustering of "single-islets". a** "Single-islet" UMAP cluster annotation comprising all donor islets and stratification according to PT/PH regions (subcluster IV-E is sample-biased and excluded from further consideration). **b**, **c** UMAP clustering of all islets with color gradients indicating relative staining area percentages for individual endocrine hormones (min-max: endocrine/CHGA: 0–100%; ProINS: 0–90%; INS: 0–50%; IAPP: 0–40%; ProGCG: 0–60%; GCG: 0–30%; SST: 0–20%; PPY: 0–100%). **d** Histograms pertain to indicated UMAP clusters and display islet feature distributions combined from all donor PT/PH sections;

adjacent heatmaps stratify islet size and relative hormone staining areas (CHGA, ProINS, INS, GCG) across T1D stage, individual donors (listed in order of increasing age within each group), islet cluster affiliation, and PT/PH regions. Note that not all donors have islets populating each cluster and we further omitted values if clusters contained <3 islets or <2 donors; as in Fig. 2, selected CHGA (Ctrl 6162) and INS (Ctrl 6162, AAb 6450) data were also excluded (missing/excluded values rendered in gray). **e** Summary of distinctive islet properties across UMAP clusters.

the trend towards larger islets in the PT must be grounded in regionally different cluster abundances. Indeed, the PH of Ctrl donors contains a prominent contingent of small cluster IV islets (median diameter: 44 µm) but combined cluster I-III ICIs have the same median size as PT islets (64 µm diameter). A similar ranking of islet circularities (cluster

I > II - III > IV/V-A/BC) suggests that cluster I islets, due to their higher circularity and general preponderance in the non-diabetic pancreas (see below), may well have contributed to the historic notion of a "standard islet"[78], and few differences in circularity (Fig. S5i) and even endocrine properties (Fig. S5j) are observed for PT and PH islets within

the same clusters. Also, note the differential ProINS and IAPP content for clustered ICIs (I > II > III); absence of alpha/beta cells and low CHGA but high PPY expression in cluster IV islets; and elevated ProGCG/GCG expression levels of cluster V islets in conjunction with slightly increased SST content (Figs. 3d, S4j and S5b, d). In agreement with the idea that INS may exert paracrine trophic effects, islet cell densities in clusters I/II exhibit an inverse correlation with beta cell content across disease progression while IDIs in PH cluster IV and the T1D-associated clusters V-A/BC are ~35% denser than cluster I/II/III ICIs (Figs. 3e, S4j and S6a). A succinct overview of these distinctive cluster properties is featured in Fig. 3e.

While UMAP analyses, by design, foreground discriminating cluster features, a comparison of islet properties within individual clusters across T1D stages also is instructive: first, in transition from Ctrl to T1DS stage, ProINS and IAPP but less so INS content is progressively reduced in cluster I, II and III ICIs suggesting that a tissue-wide modulation of ProINS and IAPP expression reports T1D pathogenetic processes with particular sensitivity (Fig. S5f, h; also cf., Fig. 2q). Second, the opposing trajectories for islet size increase vs. circularity decrease in cluster II ICIs and cluster V-A/BC IDIs can serve as a histopathological correlate for the dynamics of disease progression (Fig. S5e, g). Taking into account the reciprocal relation of islet size and circularity (Fig. 2o) and the disease course-associated ICI to IDI transition (Fig. 2a), our observations are in fact consistent with islet size-dependent T1D kinetics: an early disappearance of smaller/more circular cluster II ICIs (turning into smaller/more circular V-A/BC IDIs while leaving behind larger/less circular cluster II ICIs), and a later appearance of larger/less circular cluster V-A/BC IDIs that are derived from larger/less circular cluster II ICIs (Fig. S5e, g).

## Islet cluster size redistribution as an early hallmark of T1D development

Arguably most relevant is the profound redistribution of relative cluster magnitudes with disease progression (Fig. 4a). Heatmap (Fig. 4b) and scatter plot (Fig. S6b) displays demonstrate that cluster I/II islets account for the majority of islets in Ctrl pancreata (PT: cluster I ~ 70%, II ~ 25%; PH: cluster I ~ 48%, II ~ 17%; also III ~ 25%; IV ~ 10%). In AAb donors, this distribution is markedly altered due to a 39-45% reduction of cluster I islets accompanied by a compensatory 45–87% increase of cluster II islets (Fig. 4c); this conclusion also applies to the individual collapse of subclusters I-A/B that are primarily distinguished by the presence/absence of delta cells (Fig. S6c), and cluster III islets in the PH including both SST+ and SSTneg subsets (i.e., cluster III-A/C reduction and cluster III-B/D increase) (Fig. 4d, e) therefore denoting the same fate for ICIs regardless of SST content.

In transition to the AAb stage, cluster V islets can be found in some donors, and a progressive deterioration of cluster I/II magnitude in T1DS donors is accompanied by a robust increase of cluster V islets. In the PH, this pattern coincides with a relative growth of cluster IV (which constitutively lacks alpha/beta cells), including all of its SST+ and SSTneg subclusters (Figs. 4c and S6d). The reconfiguration of cluster sizes culminates in T1DL cases where IDIs in cluster V (PT) or clusters IV/V (PH) practically eclipse a minute fraction of residual cluster II/III ICIs (Fig. 4a–e). Throughout T1D progression, distinctive differences between PT and PH cluster abundance (clusters I/II/V: PT > PH; clusters III/IV: PH > PT) remain broadly intact (Fig. S6e), indicating that despite regional differences, T1D pathogenesis proceeds in a somewhat synchronized fashion throughout the entire pancreas.

Translating the dynamic regulation of relative islet cluster magnitudes into the context of whole tissue sections illustrates the gradual progression of cluster-stratified islet densities with T1D stages, especially the significant reduction of AAb cluster I islet densities in the PT or of combined cluster I/III islet densities in the PH (Fig. 4f). Furthermore, the conspicuous growth of PH cluster IV with disease progression (Fig. 4c, f) is readily explained by cluster III ICIs that, after loss of beta cells, segregate together with cluster IV IDIs, thereby raising average islet size and lowering circularity (Fig. S5g). Lastly, a donor-specific assessment of cluster frequency distributions (Fig. 4b) in conjunction with clinical donor parameters (Supplementary Data 1) permits the identification of several outliers (Ctrl 6454, AAb 6310, AAb 6450, T1DS 6405) and associated considerations, as detailed in the legend to Fig. S6e.

## Insulitis and insulitic islet properties

To quantify immune cell abundance, distribution, and islet association, we first counted CD45+ immune cells localized within (intra-islet) or immediately surrounding (peri-islet) individual islets (cf., Figs. 1g and S1c) and assessed instances of insulitis (≥15 CD45+ cells associated with ≥3 islets[79]). As shown in Fig. 5a, automated insulitis diagnoses (0/7 Ctrl, 3/6 AAb, 6/8 T1DS, 1/4 T1DL donors) are in overall concordance with prior manual determination by nPOD pathologists (Supplementary Data 1). Insulitis is an expectedly rare occurrence in AAb donors (PT/PH: 0.8/0.6% of all islets), constitutes an infrequent event in T1DS (PT/PH: 3.2/1.3%), preferentially affects ICIs, and recedes to levels comparable to Ctrl donors in the T1DL stage[12] (Fig. 5b). According to our UMAP cluster stratification, insulitis distributions are preferentially restricted to cluster II where 14% (PT) to 18% (PH) of islets are insulitic in T1DS cases; such lesions are less pronounced in cluster V-A, very rare in clusters III/IV, and absent in clusters I/V-BC (Fig. 5c, d). Despite lower insulitis frequencies in AAb donors, islet-associated CD45+ counts in cluster II are higher than for Ctrl individuals comparing pooled donor groups, and an average of 2-3 CD45+ cells per islet in cluster V-A of AAb, T1DS and T1DL donors constitutes a residual presence of immune cells no longer engaging beta cells (Fig. 5d).

Since cluster II islets histologically appear to constitute the principal target of autoimmune attack, we next compared features of cluster II insulitic vs. non-insulitic islets in AAb and T1DS donors. Insulitic islets are notably larger with a modest reduction in cellular density, and they tend towards reduced circularity, solidity and beta cell content but increased alpha-cell proportions (Fig. 5e). However, since these features track with islet size (cf., Fig. S4a–h), there appear to be no fundamental differences between most insulitic and non-insulitic islet properties. Altogether, our findings demonstrate the utility of automated image analysis strategies for expedient insulitis diagnosis and reveal a largely normal architectural and endocrine composition of insulitic islets.

## Islet-associated immune cell burden: from physiological to pathological

In aggregate, i.e., across all disease stages and pancreas regions, ~52% of islet-associated immune cells are found in cluster II, ~8% in cluster III, ~13% in cluster IV, and ~26% in cluster V-A. Notably, ~40% of CD45+ cells therefore associate with cluster IV and V-A islets despite their lack of beta cells (frequencies of IDIs with elevated CD45+ cell counts or insulitis reportedly range from ~1–8%, depending on varying insulitis definitions, islet pools selected for analysis and T1D stage[12,28,33]; in contrast, our estimates emphasize overall immune cell-islet associations beyond IDI insulitis which in our comprehensive islet captures is consistently <1%; Fig. 5b). Since the use of absolute islet-associated CD45+ cell numbers in the consensus insulitis definition does not clearly account for islet size[79], some investigators have adopted strategies for islet size correction[25,28,80]. We employed a similar approach by quantifying islet-associated CD45+ frequencies, namely the numbers of CD45+ cells per nuclei in islets and peri-islet regions. Consistent with the overall cluster affiliation of islet-associated CD45+ cells, islets with increased CD45+ cell frequencies primarily localize to clusters II/V-A and, to a lesser extent, also clusters III/IV (Fig. 6a, b). While these findings echo the distribution of insulitic islets, they further demonstrate a physiological component of immune cell infiltration since

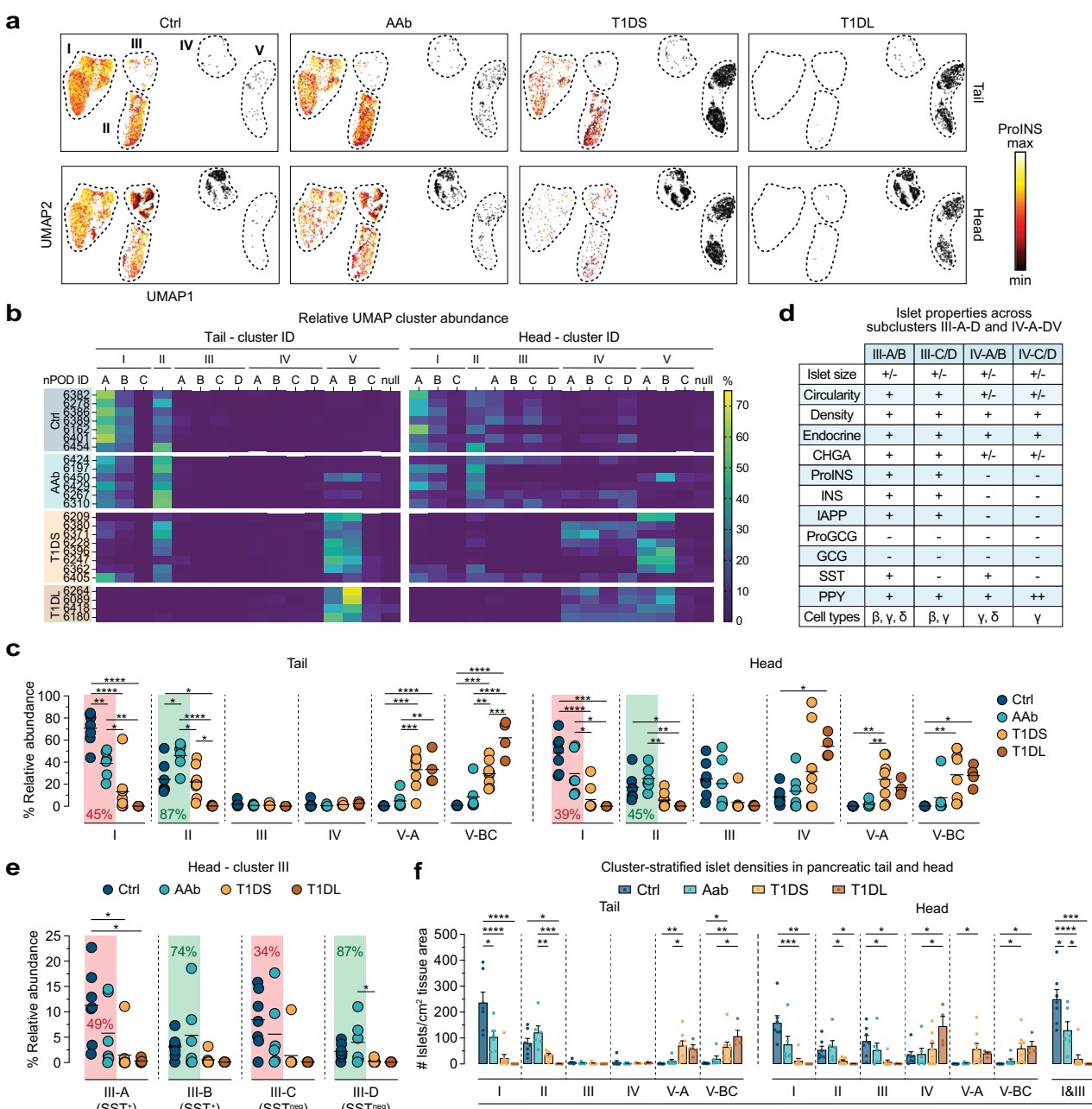

**Fig. 4 | Redistribution of relative UMAP cluster sizes as a distinctive feature of early T1D development. a** UMAP cluster display of all islets as a function of pancreas region and disease stage; the color gradient pertains to relative ProINS expression levels (min-max: 0–90%). **b** Heatmaps stratifying relative UMAP cluster sizes across T1D stages (individual donors listed in order of increasing age within each donor group), cluster affiliation (null: unclustered), and PT/PH regions. **c** Relative cluster magnitudes in PT/PH across T1D stages; Ctrl *vs.* AAb donors: note the relative decline of cluster I size means (red background) and accompanying increase of cluster II size means (green background). **d** Summary of distinctive islet properties in PH subclusters III-A-D/IV-A-D (a further distinction of the four

subcluster pairs according to differential immune cell burden is detailed in Fig. 6a). **e** Relative magnitudes of subclusters III-A-D in PH across T1D disease stages.
**f** Densities of cluster-stratified islets in parenchyma of PT (left) and PH (right; the far-right plot features combined cluster I/III islets since the redistribution dynamics for these clusters are comparable, *cf.* **c**, **e**). Colored circles in scatter plots panels (**c**, **e**) display mean values derived from individual donors and black horizontal bars are donor group means with $n = 7$ Ctrl, 6 AAb, 8 T1DS and 4 T1DL donors; data in panel (**f**) are scatter and mean±SE. Statistical analyses in (**c**, **e**, **f**) were conducted with ordinary one-way ANOVA and Tukey's multiple comparisons test (*$p < 0.05$, **$p < 0.01$, ***$p < 0.001$, ****$p < 0.0001$).

cluster II islets in Ctrl donors, accounting for ~17% (PH) to ~25% (PT) of all pancreatic islets, on average harbor ~1.7% CD45$^+$ cells in the peri-islet region alone. This fraction increases by ~3-fold in T1DS donors (Figs. 6c and S6f), even if T1DS 6209, a young child with notably high peri-islet infiltrates previously considered a more aggressive endotype 1 T1D case[81,82], is excluded from analyses. Intra-islet CD45$^+$ frequencies tend to be lower but are also elevated 2.4–5.2-fold in the T1DS group, and similar considerations apply to PH cluster III islets

(Figs. 6c and S6f). In contrast, cluster IV/V-A IDIs follow a different pattern: higher peri-islet CD45$^+$ cell frequencies in AAb donors (PT: 3.5%, PH: 4.2%) decline to ~2.1% in T1DL cases with less pronounced trends also observed for intra-islet CD45$^+$ cells (Figs. 6c and S6f). The fact that ratios of peri-islet to intra-islet CD45$^+$ frequencies tend to rise in cluster II ICIs and fall in cluster V-A IDIs across T1D stages (Fig. S6g) may provide clues to better untangle the temporal progression of immune cell-mediated beta cell destruction.

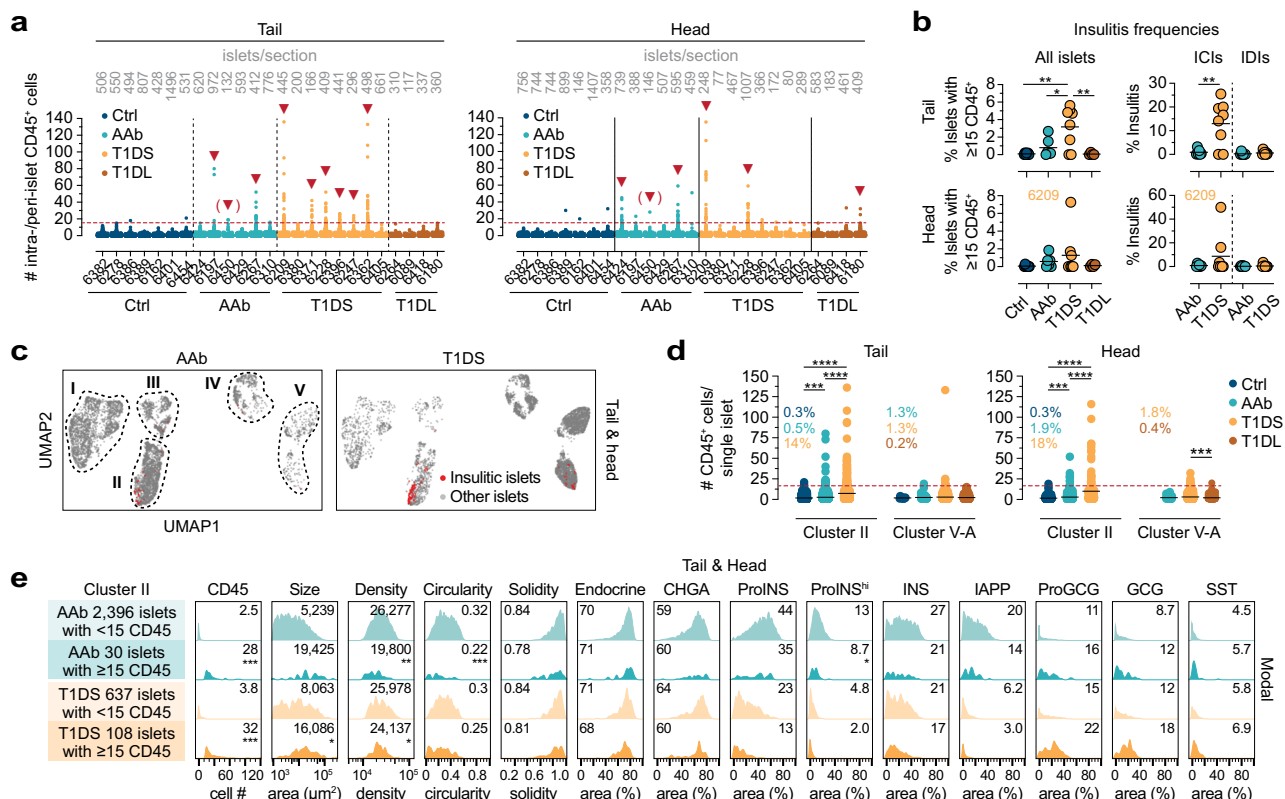

**Fig. 5 | Insulitis and insulitic islet properties. a** Total numbers of islet-associated CD45+ cells (*i.e.*, intra- and peri-islet) for every single PT (left) and PH (right) islet from all donors (listed in order of increasing age within each group; gray values on top are numbers of islets captured for each donor). The dashed red line indicates the insulitis threshold of ≥15 islet-associated CD45+ cells, and red arrows highlight donors with ≥3 islets meeting the insulitis definition (AAb 6450: insulitis in two PT and one PH islet). **b** Insulitis frequencies for all islets and ICI *vs.* IDI subsets (colored circles: mean values derived from individual donors, black horizontal bars: donor group means). **c** UMAPs depicting cluster localization of insulitic islets in AAb and T1DS donors. **d** Absolute CD45+ cell numbers associated with individual islets in clusters II ICIs and V-A IDIs of respective donor groups (each symbol represents an

islet; dashed red line: insulitis threshold; color-coded values are percentages of insulitic islets in respective PT/PH clusters). **e** Properties of cluster II insulitic *vs.* other islets in AAb and T1DS donors (combined PT/PH islets from all AAb or T1DS donors; values are means, or medians for islet size/area; density: islet cells/mm² islet area); asterisks indicate statistical differences calculated for respective insulitic *vs.* non-insulitic islet properties from individual PT/PH donor sections. Statistical analyses were conducted with ordinary one-way ANOVA and Tukey's multiple comparisons test (**b**/left, **d**), unpaired two-tailed Student's *t* test (**b**/right), or paired two-tailed Student's *t* test (**d**) with n = 7 Ctrl, 6 AAb, 8 T1DS and 4 T1DL donors (*p < 0.05, **p < 0.01, ***p < 0.001, ****p < 0.0001).

Lastly, consideration of the overall islet-associated immune cell burden reveals a striking ~2.5-fold increase of CD45+ cell numbers and densities already at the AAb stage (Fig. 6d). Though neither significant nor further increased with T1D onset, the proportional extent of this increase mirrors the relative enlargement of the islet-reactive CD8+ T cell pool in AAb pancreata[83]. Taking into account corresponding beta cell-specific CD8+ T cell frequencies in peripheral blood[83], early immune cell recruitment to pancreatic islets may thus be largely stochastic before enrichment of specific CD8+ T cells becomes discernible at T1D onset. In the T1DL stage, islet infiltration has for the most part subsided (Fig. 6d). Altogether, these observations combine into a specific histopathological correlate for the AAb stage: the frequency of islets associated with ≥1 CD45+ cell constitutes a straightforward metric that readily distinguishes Ctrl (25%) from both AAb and T1DS (45–55%) donors and therefore captures a significant immune cell recruitment to islets in stage 1/2 T1D (Fig. 6e).

**Compromised islet composition in the absence of immune cells**
A donor group-specific correlation between CD45+ burden and islet properties in cluster II, the apparent histopathological locus of CD45+ cell infiltration, is confounded by the overall contingency of islet features on islet size (Figs. 2m–q and S4a–h), including islet-associated CD45+ cell numbers and frequencies (Fig. S6h); instead, we compared size-binned cluster II islets across donor groups. Here, an increase of

CD45+ numbers with disease progression appears to have little bearing on islet architecture or alpha and delta cell abundance (Figs. 6f and S6i). In contrast, ProINS and IAPP, but not INS, content is markedly reduced in T1DS *vs.* Ctrl and AAb donors, especially in smaller islets (Fig. 6f). Even more striking is the observation that similar though attenuated patterns are also recorded for cluster I islets, which do not harbor any CD45+ cells (Figs. 6g and S6i). Cluster I islet impairment may arise following exhaustion of residual beta cells tasked with overwhelming metabolic demands[52,84] or as a consequence of beta cell-targeted autoimmune attack[85] not captured here due to its dynamic nature and/or the restricted plane of histological analysis; in all likelihood, both scenarios apply to varying degrees to different cluster I and other ICIs[6].

Collectively, our cluster I/II islet evaluation indicates the lack of a straightforward correlation between CD45+ cells and islet composition, suggesting that T1D pathogenesis is at once highly dynamic (*i.e.*, CD45+ cells may "come and go" before complete beta cell destruction is achieved), initially targets islets below average size (compounding T1D symptomatology since smaller islets are capable of comparatively greater INS secretion[86,87]), and appears broadly synchronized since it affects the majority of islets (*i.e.*, ICIs present with reduced ProINS/IAPP expression even in the histological absence of immune cells). The utility of the insulitis concept and its proposed amendments notwithstanding[88], considerations of the entire range of immune cell

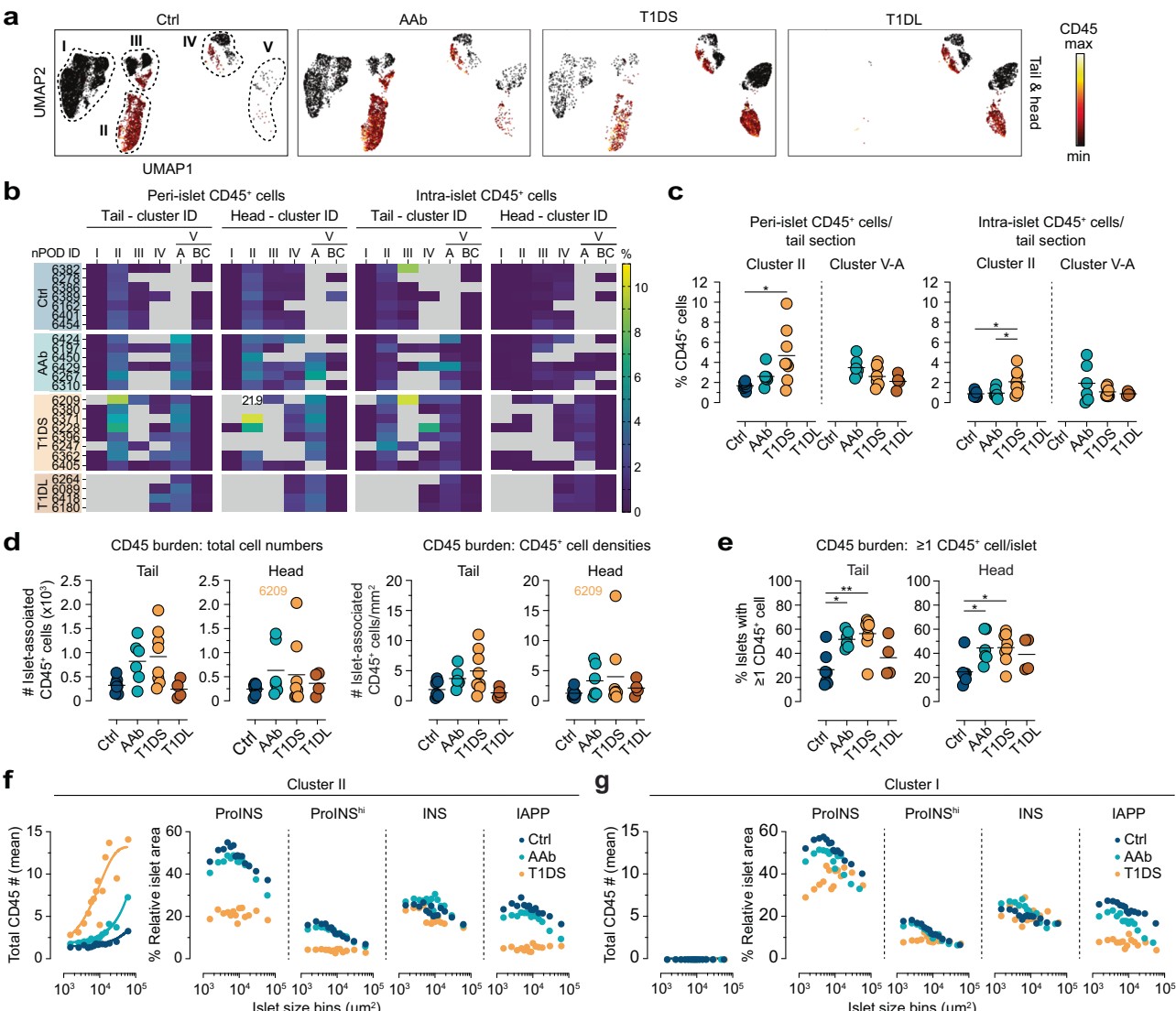

**Fig. 6 | Islet-associated immune cell burden. a** UMAPs of combined peri- and intra-islet CD45⁺ cell frequencies (CD45 color gradient [min-max]: 0-10%) across T1D stages in PT/PH. **b** Heatmaps of average peri-/intra-islet CD45⁺ cell frequencies stratified across donor group, individual donors listed in order of increasing age within each donor group, cluster affiliation, and PT/PH regions (note out-of-range value for T1DS 6209 cluster II peri-islet CD45⁺ cell frequency; missing/excluded values in gray). **c** Peri- and intra-islet CD45⁺ cell frequencies in PT clusters II/V-A (note that T1DL and Ctrl donors lack cluster II and V-A islets, respectively). **d** Total number of islet-associated CD45⁺ cells in tissue sections (left), and islet-associated CD45⁺ cell densities (CD45 numbers/mm² peri-islet/intra-islet area; right).

**e** Frequencies of islets with ≥1 associated CD45⁺ cell(s) across disease stages in PT/PH. **f** Combined PT/PH cluster II islets stratified across donor groups (Ctrl, AAb, T1DS) were binned according to islet size for display of associated CD45⁺ burden and ProINS/INS/IAPP expression (small colored circles are islet size bin-specific donor group means; lines in left panel are one-phase association curve fits with $R^2 > 0.75$). **g** Same as panel (**f**) but for cluster I islets. Statistical analyses in panels (**c–e**) were conducted with $n = 7$ Ctrl, 6 AAb, 8 T1DS and 4 T1DL donors using ordinary one-way ANOVA and Tukey's multiple comparisons test (colored circles: mean values derived from individual donors; black horizontal bars: donor group means; *$p < 0.05$, **$p < 0.01$).

---

associations with pancreatic islets are therefore necessary to clarify progression of T1D autoimmune processes.

**Spatial distribution of islets throughout T1D pathogenesis**

We employed three complementary approaches to assess the spatial distribution of islets in pancreatic tissue sections: modified Ripley's K function, Delaunay triangulation, and direct visualization of cluster-associated islet localities. Ripley's K function[89,90] is a descriptive statistic for detecting deviations from spatial homogeneity, here the density enrichment of islets in contrast to their random distribution patterns (Fig. 7a and Methods). Density enrichments, *i.e.*, islets presenting as non-randomly distributed "aggregates", are observed in the PT of all donor groups across radial distances from

~1000–10,000 µm yet their extent is notably different (Fig. 7b): in Ctrl donors, a modest ~1.2-fold density enrichment for islets in the ~600–1500 µm range is accompanied by a convergence toward random distributions for greater radii (Fig. 7b); thus, a relatively even distribution of islets throughout the PT readily aligns with recent 3D interrogations of non-diabetic pancreata[35]. While near identical patterns are recorded for AAb donors, more pronounced islet density enrichments occur in T1DS (~1.9-fold, peaking at 800 µm) and T1DL (~2.5-fold, peaking at 600 µm) (Figs. 7b and S7a). Similar observations also pertain to the PH but the unique anatomy of the uncinate process (see below) is responsible for overall greater density enrichments in all donor groups (Figs. 7c and S7a). Importantly, since islets do not change their anatomic locations, their density

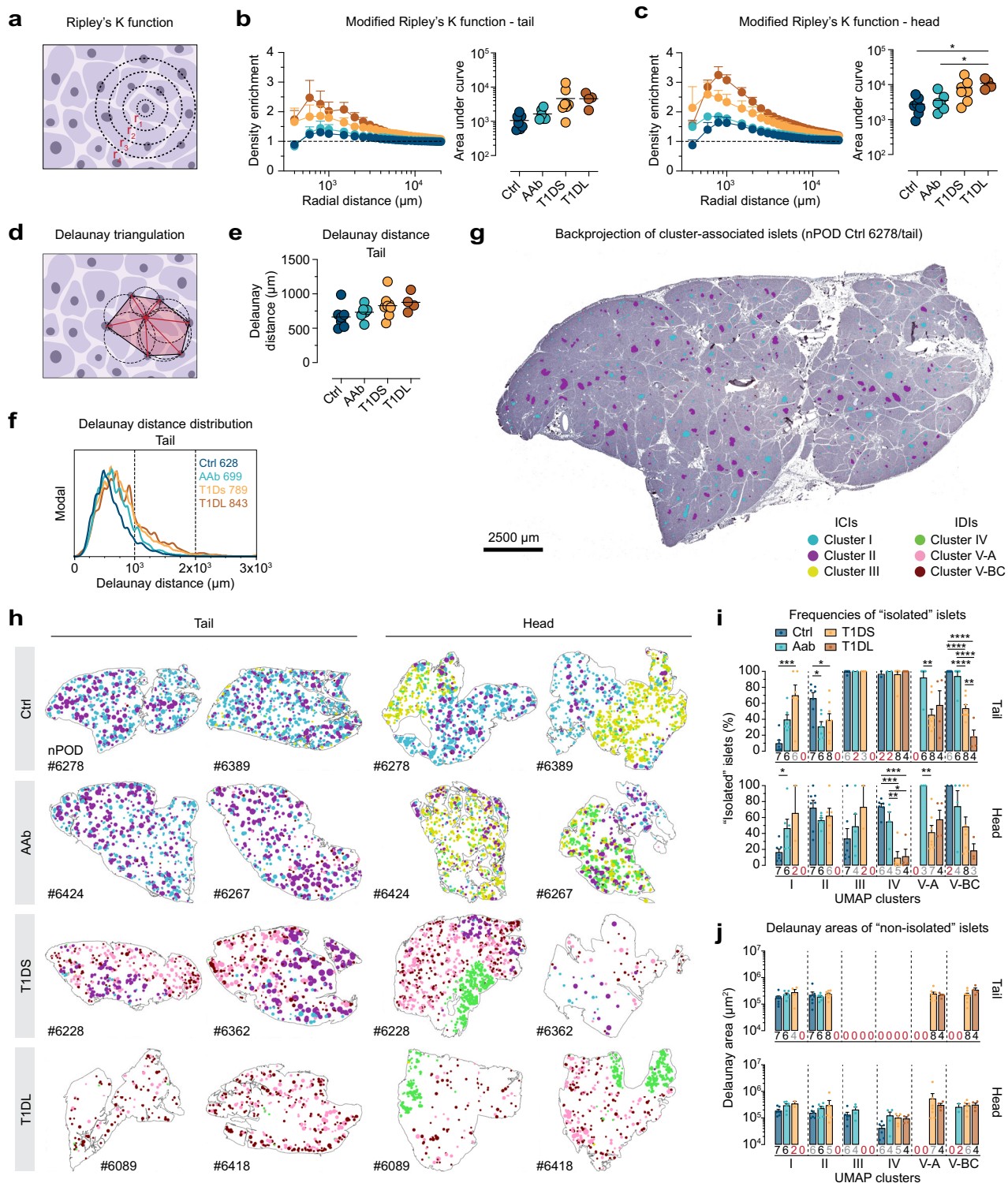

**h** (Tail / Head panels) with rows Ctrl, AAb, T1DS, T1DL; nPOD #6278, #6389, #6424, #6267, #6228, #6362, #6089, #6418.

**ICIs** — Cluster I, Cluster II, Cluster III
**IDIs** — Cluster IV, Cluster V-A, Cluster V-BC

enrichments need to be considered in a relative fashion: because of islet loss at larger distances and a resulting "re-definition" for random distributions at lower islet densities, increasing density enrichments over disease course reflect a progressive reduction of tissue regions with unperturbed islet densities (Fig. 7b, c). Altogether, these observations are consistent with the overall decrease of islet densities, cumulative areas, and mass (Figs. 1i and S2h), and they are further confirmed by fractal dimension analyses[91] as illustrated in Fig. S7b.

**Progressive changes to islet neighborhoods**

Delaunay triangulation organizes a set of points (*i.e.*, islets) in a plane (*i.e.*, tissue sections) into triangles whose circumcircles do not contain any of the points[92]; thus, mean Delaunay distances and triangle areas can serve as a more local metric for average adjacency between individual islets and their proximate neighbors (Fig. 7d). Here, Delaunay distances for PT islets increase by ~30% from Ctrl to T1DL stage and display a similar trajectory in the PH (Figs. 7e and S7c). At the same time, mean Delaunay distances are normally distributed in Ctrl

**Fig. 7 | Spatial distribution of islets in the pancreas across the course of T1D pathogenesis. a** Ripley's K function: input values are radial distances between each islet and expanding concentric circles serving as bins to quantify surrounding islets. **b** Left: donor group-stratified islet density enrichments in PT as a function of radial islet distance-bins from $4 \times 10^2 - 2 \times 10^4$ mm (data are mean±SE); horizontal line indicates random islet distribution, values > 1 denote islet density enrichments. Right: areas under Ripley curves. **c** Data display for PH islets as in panel b (T1DS 6380 excluded due to scarce islets, *cf*. Fig. S8a). **d** Delaunay triangulation of islets and associated circumcircles. The average length of six red triangle sides/lines connecting the central with neighboring islets is the mean Delaunay distance; average triangle area is mean Delaunay area. **e** Mean Delaunay distances between islets across T1D progression in PT. **f** Mean Delaunay distance distribution according to T1D stage. **g** In situ projection of islets color-coded by cluster onto representative PT section. **h** Pancreatic tissue section outlines populated with islets in their original location color-coded by UMAP cluster affiliation (islets rendered circular/enlarged with size differences preserved; color legend in **g**). **i**, **j** Delaunay triangulation was performed separately for islets in each cluster; islets with <2 islets within an 8 mm distance are "isolated islets". Note that some clusters lack islets (*cf*., Fig. 4f) and we also excluded data from calculations of fractions/areas if <3 islets or <2 donors were present. The resultant number of donors represented in each cluster and disease stage is indicated by color-coded values above x-axes (black: all donors; gray: reduced number of donors; red: instances with only 2 donors excluded from statistical analyses, or no donors [absent/excluded data]; data are scatter and mean±SE). Colored circles in scatter plots (**b**/right, **c**/right, **e**) display mean values derived from individual donors (black horizontal bars: donor group means) with n = 7 Ctrl, 6 AAb, 8 T1DS and 4 T1DL donors. Statistical analyses in (**b, c, e, i, j**) conducted with ordinary one-way ANOVA and Tukey's multiple comparisons test or two-tailed Student's *t* test, as applicable (*$p < 0.05$, **$p < 0.01$, ***$p < 0.001$, ****$p < 0.0001$).

pancreata with a slight extension of the right distribution tail representing islets that are more distant to each other; with T1D progression, the respective right distribution tails increase in particular in the PT (Figs. 7f and S7c), and further alterations can be discerned by a cluster-level evaluation of Delaunay distances as detailed in Fig. S7d–g. Collectively, our observations support the notion that regionalized patterns of T1D histopathology, rendered apparent by ICI distributions observed in common INS-stained tissue sections[93], are underpinned by a tissue-wide islet depletion, the extent of which is partially obscured by its fundamentally dispersed nature.

### Cluster-specific islet distributions and the dynamics of T1D progression

To visualize the above relations more directly, we projected cluster-stratified islets back onto their original tissue section locations (Figs. 7g and S7h), revealing patterns unique to specific T1D stages: in the Ctrl PT, a relatively uniform distribution of cluster I islets is accompanied by a sparser presence of cluster II islets, a balance that in AAb donors shifts to visibly fewer cluster I islets. This pattern changes more decisively in T1DS cases where the appearance of predominantly cluster V-A islets crowds out remaining cluster II islet foci; this process is completed in T1DL with the prominent emergence of cluster V-BC islets that often appear to populate the tissue area periphery (Figs. 7h and S8a). In the PH, a similar progression pertains to the redistribution of cluster I/II/V islets, yet the most distinctive aspect is a striking regionalization of cluster III/IV islets that represent the uncinate process (Figs. 7h and S8a).

We employed islet cluster-specific Delaunay triangulations to quantify these impressions. Here, "isolated islets" were defined as islets for which, due to <2 islets from the same cluster within an 8 mm distance, Delaunay triangulation was not possible. As cluster I magnitudes collapse from Ctrl to T1DS stage (Fig. 4f), the fraction of "isolated islets" increases (Fig. 7i). At the same time, "isolated islet" frequencies in cluster II decline somewhat, reflecting a relative condensation of residual cluster islands that are in the process of disappearing with T1D onset (Figs. 4f and 7h, i). The sparse cluster III/IV islets in the PT are practically all isolated and remain so throughout all disease stages. In contrast, the more abundant cluster III ICIs in the PH become increasingly isolated as their transition to IDI status following beta cell loss leads to consolidation with cluster IV islets (*cf*., Fig. 4d, f); in turn, the resultant growth of cluster IV is accompanied by a steep decline of "isolated islets" therein (Fig. 7h, i). "Isolated islets" in clusters V-A/BC dominate at first appearance in the Ctrl and/or AAb stage, decline in T1DS with expansion of respective cluster magnitudes (Fig. 4f), and for cluster V-BC are further reduced in T1DL (Fig. 7h, i).

The combined dynamics of these changes indicate that cluster V-A represents a transitional and V-BC a terminal fate. Complementary quantifications of Delaunay areas for "non-isolated islets" reveal trends that partially mirror the "isolated islet dynamics" (*e.g.*, increasing Delaunay areas for cluster I islets), yet differences across disease stages remain non-significant (Fig. 7j). Altogether, these observations reinforce the notion that expanding areas of islet loss, increasingly populated by growing "diabetic" cluster V islands, encroach on progressively shrinking cluster I/II/III islands.

### Insulitis and immune cell burden in spatial context

Finally, we leveraged the "back-projection" of islet subsets to visualize the neighborhood context of insulitic lesions. Using a color gradient to display the numbers of islet-associated immune cells, the resultant images demonstrate that rare instances of insulitis in AAb and T1DS donors often are the foci of larger neighborhoods with an overall increased CD45+ cell burden, a pattern that tends to be more pronounced for the PT than PH (Figs. 8a and S8b). And lastly, our spatial data visualization also clarifies the distinctive pathology of previously discussed outlier cases and is detailed in Fig. S8b.

## Discussion

This study combines multiplexed brightfield IHC of human pancreatic tissue sections, high-magnification whole-slide imaging, digital pathology, and development of a semi-automated image analysis pipeline implemented in the open-source pathology platform QuPath[42,43] to trace the evolution of T1D development and progression. Integration of these modalities readily confirms the central tenets of pancreatic T1D histopathology and reveals multiple unique aspects about the dynamic constitution of the pancreas in health and T1D disease, including a broad contingency of islet properties on their size (Figs. 2m–q and S4a–h) as well as an essentially identical endocrine organization across pancreas regions (summarized in Fig. S5i, j) that locates residual differences specifically to the uncinate process in the PH (Fig 7h and S8a; without that distinction, average values for some PH *vs*. PT islet properties may be different as shown previously[57,66]).

Notably, we define a series of distinct histopathological correlates for the stage 1/2 pancreas (enhanced immune cell burden as reflected by higher frequencies of islets with ≥1 associated immune cell[s] [Fig. 6e]; early targeting of small, including "GCG-deficient" islets [Figs. 2c, g, h, q, 6f, g, S4a–f and S5a, c, e, g]; relative reduction of islet and cumulative ProINS/IAPP areas [Figs. 2q and S2i]; and a pronounced collapse of islet cluster I magnitude [Fig. 4]) that position AAb donors on the cusp of developing the very histopathological hallmarks that are distinctive for T1D. With onset of clinical T1D, several histopathological signatures emerge that collectively suggest a tissue-wide distributed rather than strictly localized disease process. Initial support for this contention is provided by alterations of islet architectural features in clinical T1D that are somewhat independent of islet size and in T1DS only spare larger islets (Figs. 2n, o and S4a, c, e, g). We further demonstrate an absence of a direct correlation between islet composition and associated immune cell abundance (predominantly normal configuration of insulitic islets, their comparatively larger size and

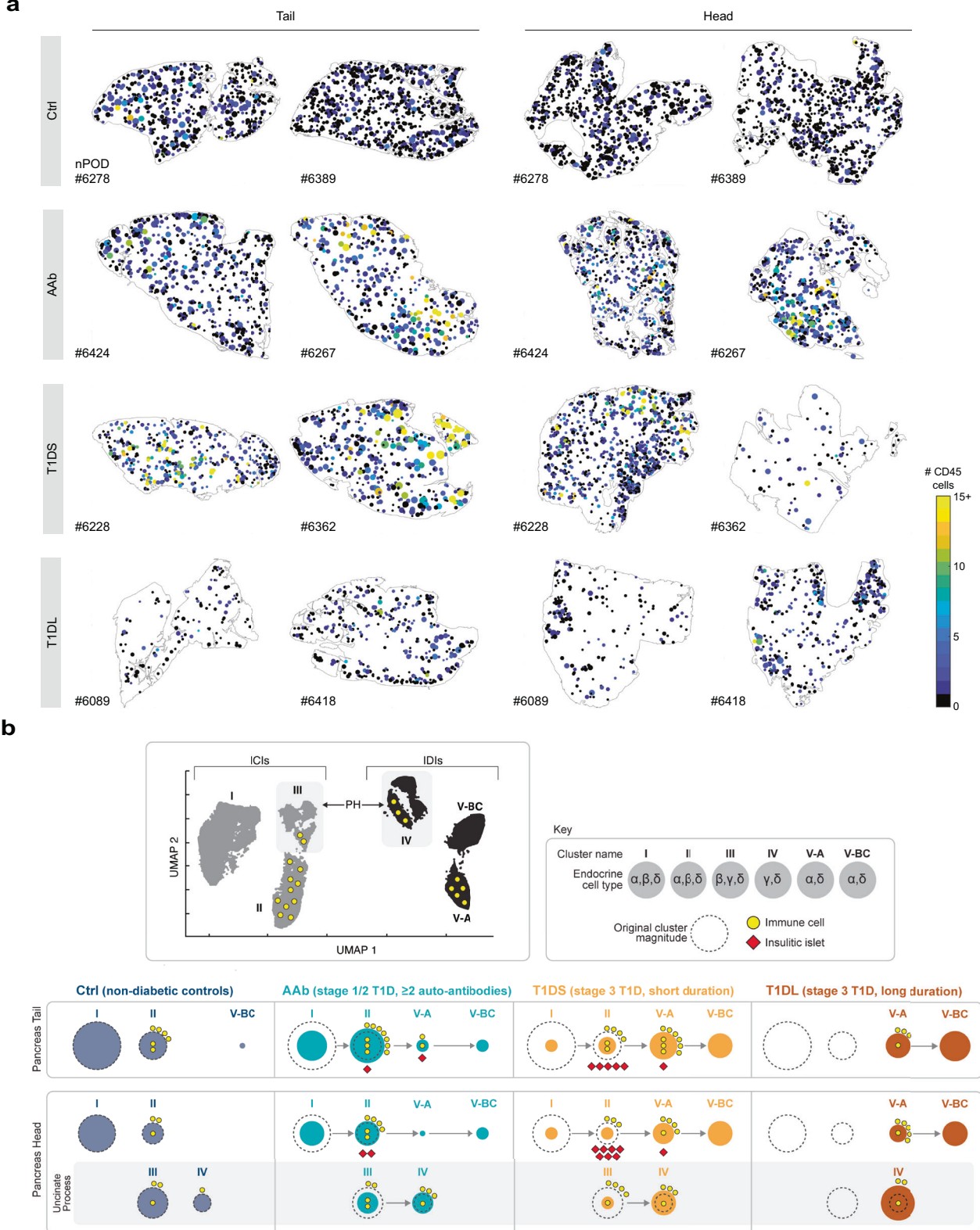

**Fig. 8 | Spatial context for islet-associated immune cell burden and model summary. a** Pancreatic tissue section outlines (regions, T1D stage and nPOD donor IDs indicated) populated with islets color-coded according to the numbers (0–15+) of associated CD45⁺ cells (islets are rendered circular and enlarged with relative size differences preserved). **b** Summary model: islets are allocated to UMAP clusters I–V-BC according to geometric properties, endocrine cell content, association with immune cells, and Delaunay area. Top left: major cluster properties of ICIs (gray) and IDIs (black); top right: legend. Bottom: T1D progression in the PT and PH (divided into non-uncinate and uncinate process regions). The size of colored circles represents the relative magnitude of respective islet clusters across T1D stages; their changing sizes across disease progression is indicated by arrows between adjacent circles; the dashed perimeter lines indicate the original magnitude of clusters I, II, III and IV in Ctrl donors and serve as a visual reference for cluster size changes with disease progression (a further net reduction of islet cluster magnitudes due to loss of pancreas weight is not captured here; also note that cluster III islets, upon losing beta-cells, become re-classified as cluster IV islets).

associated lesser cellular density a consequence of the nature of the insulitis definition itself [Fig. 5e]; and decreased ProINS/IAPP content even in smaller T1DS ICIs without observable CD45$^+$ cells [Fig. 6g]) emphasizing the importance to consider islets and their associated immune cell burden beyond those affected by insulitis.

Our distinction of UMAP-stratified islet clusters provides the foundation for a reconstruction of plausible histopathological dynamics underlying T1D development/progression (Fig. 8b), and similar patterns of beta cell reduction within clusters I-III across disease course (Fig. S5f, h) emphasize the distributed nature of T1D pathogenesis. Furthermore, our analyses of spatial islet distributions foreground the "negative spaces" of T1D histopathology, namely the apparent loss of whole islets from expanding pancreas regions that emerge through a progressive contraction of residual areas with normal islet density (Figs. 7, 8a, S7 and S8a, b), thus consolidating the concept of a multifocal and more synchronized origin and progression of T1D pathology (note that these considerations remain applicable despite the possibility of a generalized reduction of inter-islet distances due to exocrine pancreas shrinkage). Altogether, we therefore propose a revised natural history for T1D (Fig. 8b) consistent with the recent hypothesis advanced by Martino et al. that autoimmune destruction of beta cells progresses at a relatively constant rate throughout all stages of T1D[94]. The specific histopathological correlates identified here may be leveraged for more targeted investigative tasks seeking to elucidate the autoimmune pathogenesis of T1D and to promote pathology-informed considerations especially of early-stage interventional modalities.

Our study offers both additional analytical opportunities and exhibits several limitations. While "single-islet" UMAP clustering aids in better resolving histopathological properties and processes, our specific input variables may not be available for other studies. We therefore provide a simple key that permits an approximate clustering of islets based on standard three-parameter stains (INS, GCG, CD45; PPY visualization may be added to distinguish cluster III/IV islets in the PH; Fig. S8c) and thus can be readily applied to suitable archival pancreas images. While we have excluded endocrine objects <1000 μm$^2$ for the present project, their fates are gaining increased attention[95–98] and constitute a focus of ongoing investigations. The refined analysis modalities and distinct observations reported here are grounded in effective capture of known histopathological hallmarks of T1D progression yet will require validation in independent studies and larger cohorts. Our documentation of islet mass reduction in clinical T1D may be confounded by the possibility that individuals prone to T1D development have inherently smaller pancreata, a contention that at present, however, remains speculative[6]. Lastly, age-matching of pancreas donors is critical to our study design but implies different ages for disease onset; additional investigations should therefore include donors matched for age-of-onset to better account for age-dependent variables in T1D pathogenesis.

## Methods

### Organ procurement and experimental specimens detail

Donor organs and tissues were obtained by the Network for Pancreatic Organ donors with Diabetes (nPOD, RRID:SCR_014641) through its partnership with United States organ procurement organizations in accordance with federal guidelines for organ donation and as approved by the University of Florida (UF) Institutional Review Board, after consent for organ donation and research was obtained from donors' legal representatives. Recovered tissues were processed by the nPOD Organ Processing and Pathology Core at UF, and all samples, associated data, and metadata were de-identified in full accordance with privacy and ethical guidelines. In the present study, which constitutes Not Human Subjects Research (according to 45 CFR 46), we used formalin-fixed, paraffin-embedded (FFPE) tissue sections (5 μm) from pancreatic tail and head regions of 25 organ donors allocated to

four groups: 7 non-diabetic control donors (Ctrl), 6 autoantibody-positive (AAb) donors, 8 donors with short duration of clinical type 1 diabetes (T1DS, <2 years), and 4 donors with longer duration of clinical T1D (T1DL, 8–11 years); donor matching across the 4 groups was performed on age and gender with additional matching for demographic (ethnicity) and clinical (body mass index/BMI) parameters where possible; further details including demographic and clinical metadata are provided in Supplementary Data 1 (see also nPOD Data Portal https://portal.jdrfnpod.org/). To establish, validate and optimize staining protocols, additional pancreatic and splenic FFPE tissue sections were provided by the nPOD consortium and the ISMMS Biorepository and Pathology CoRE.

### Immunohistochemistry (MICSSS)

FFPE tissue sections (5 μm) were sequentially stained for eight islet hormones and CD45$^+$ immune cells adjusting the multiplexed immunohistochemical consecutive staining on single slide (MICSSS) method[45,46]. Iterative staining order was empirically determined for pancreatic tissues to account for differential antigen sensitivity to deterioration during consecutive MICSSS cycles. Briefly, sections were baked overnight (o/n) at 37 °C to ensure tissue adherence to slides in subsequent staining rounds. Following deparaffinization in histology-grade Xylene (Fisher Scientific #X3P-1GAL), sections were rehydrated by immersing them in a series of graded ethanol solutions (histology-grade, Fisher Scientific #A405F-1GAL) at decreasing concentrations (3x100%, 90%, 70%, and 50%) down to distilled water (5 min each) prior to heat-induced epitope retrieval (HIER) at pH6 (Citrate Buffer 20X concentrate; ThermoFisher Scientific #005000) in a 95 °C water bath for 30 min. Tissue sections were cooled down to room temperature, endogenous hydrogen peroxidase activity was quenched by incubation with PeroxAbolish solution (Biocare Medical #PXA946) for 10 min and slides were subsequently washed in Tris Buffered Saline (2x5 min TBS, 10X concentrate, Cell Signaling #12498). Non-specific background due to Fc receptor binding was blocked by incubation with DAKO serum-free protein block (Agilent #X0909) for 15 min, endogenous biotin was blocked using the DAKO Biotin-Blocking System (Agilent #X0590) according to manufacturer instructions before addition of primary antibodies as specified in Table S1. Target antigens were revealed after incubation with biotinylated F(ab')$_2$ donkey-raised secondary antibodies with minimal cross-reactivity (30 min RT; Biotin-SP AffiniPure F(ab')$_2$ Fragment, Jackson ImmunoResearch: donkey anti-guinea pig IgG #706-066-148; donkey anti-mouse IgG #715-066-151; donkey anti-rabbit IgG #711-066-152; donkey anti-goat IgG #705-066-147), horseradish peroxidase (HRP)-conjugated streptavidin (30 min RT; DAKO Agilent #P0397), and ImmPACT AMEC Red substrate (Vector Laboratories #SK-4285) as per vendor's instructions. Tissue sections were counterstained with Harris modified hematoxylin solution (Sigma #HHS32), cover-slipped with an aqueous mounting medium (DAKO Glyercgel, Agilent #C0563), whole-slide images were acquired at 40× on a NanoZoomer S60 Digital Slide Scanner (Hamamatsu) and exported as .ndpi files (for some pilot experiments, images were acquired at 20× using a Pannoramic 250 Flash II Digital Scanner [3DHistech]; cf. Fig. S1a). Slides were stored protected from light at 4 °C until further staining. For sequential staining of subsequent targets in MICSSS cycles #2-9, cover slips were carefully removed in hot water (~50 °C) and tissue sections were rinsed in distilled water, destained/dehydrated by immersing them in ethanol solutions at increasing concentration (50, 70, 100%; 3 min each) and xylene (3 × 2 min), and subsequently rehydrated through a graded ethanol series and distilled water as detailed before. Antigen retrieval at pH6 was performed for 10 min as specified above removing hematoxylin staining, endogenous and streptavidin-associated (from previous rounds of staining) hydrogen peroxidase activity, non-specific background, as well as endogenous and secondary antibody-mediated (from previous rounds of staining) biotin were blocked using

PeroxAbolish, DAKO serum-free protein block, and DAKO Biotin-blocking system as above. To enable probing for target antigens using primary antibodies raised in the same species, tissue sections were incubated with 5% mouse or rabbit IgG (Jackson ImmunoResearch, normal mouse serum #015-000-001; normal rabbit serum #011-000-001) (depending on the primary antibody used in the immediate prior staining cycle) for 1 h at RT followed by incubation with donkey-raised AffiniPure Fab Fragments (AffiniPure Fab Fragment, Jackson ImmunoResearch: donkey anti-mouse IgG #715-007-003; donkey anti-rabbit IgG #711-007-003) against the previously used primary antibody species (200 μg/mL) at 4 °C o/n. Subsequent primary antibody staining, target revelation, and image acquisition were conducted as described above. For logistical reasons, each round of MICSSS staining was performed in three batches (16–17 slides/batch, containing samples from all donor groups) stained on three consecutive days.

### QuPath analysis pipeline

**Image alignment.** Images were imported into QuPath version 0.2.3[42], with each set of 9 images of a single slide bundled into an individual project. Using the CD45 image as the base image, the other 8 project images were each aligned to it using a custom groovy script based on QuPath's Interactive Image Alignment function, incorporating affine transforms at 5 μm resolution. A threshold pixel classifier was used to detect the entire tissue area on the CD45 image, which had been acquired last and therefore exhibited all of the accumulated tissue damage from the iterative MICSSS staining process. The pancreas tissue was then transferred to each image in the project, utilizing the pre-calculated affine transform matrix.

**Islet detection.** Further image processing was performed with QuPath version 0.4.3 or 0.5.0. Stain separation vectors were optimized for each hormone staining round and consistently applied to every image with that antibody to spectrally unmix the AMEC chromogen from hematoxylin. CHGA was expected to mark all islets yet staining was notably faint in some PPY$^+$ and small GCG$^{neg}$ islets; we therefore delineated islets with a combination of six hormone stains (CHGA, ProINS, INS, GCG, SST and PPY; ProGCG and IAPP proved redundant for this task and were therefore not included to reduce computational complexity and time). On these six stains, we segmented objects with a low-resolution pixel classifier (0.88 μm), thresholding the AMEC channel to find all stained regions with an area of at least 50 μm². All objects were transferred to their related CD45 base image and then merged. The large resulting annotation was split into individual objects. Holes were filled in to create contiguous objects, and then all objects less than 1000 μm² were removed (note that there is no official consensus about a minimal islet size that distinguishes it from smaller endocrine cell clusters; rather, our choice of 1000 μm² as a suitable threshold reflects a compromise based on historical context[71,74,78]). Additionally, objects that were within 10 μm of the tissue border were removed to prevent errors from incomplete islet capture and edge staining artifacts. The proto-islets were converted to detections and redistributed to all nine stains, where their overall AMEC staining intensity was measured (mean, standard deviation, min, max, and Haralick features for texture). The measurements from each object were gathered, along with shape features, and used to train a machine-learning object classifier to remove artifacts due to dust, non-specific staining, tissue damage, or loss of focus. The classifier was trained on multiple representative images from all four donor groups. We then measured and calculated shape descriptors for each final islet, and we performed Delaunay clustering on each slide with a maximum search radius of 4000 μm (1 of 24,578 islets had no neighbors within 4 mm; this islet was assigned a Delaunay distance of 4 mm and a Delaunay area of $p$ x 16 mm²).

**Hormone staining areas and islet/endocrine cell type mass.** Islet boundaries were transferred to each of the eight hormone-stained images and a high-resolution (0.22 μm²) pixel thresholder was applied to find regions of positive staining. ProINS was the first stain in the series and therefore allowed for reliable differentiation of darker staining, lighter staining, and background. As CHGA is expected to be in nearly all endocrine cells, we captured all staining areas including lighter stained sections. For all other hormones, we only captured darker staining areas to avoid inclusion of background signal. The detections representing the stained area per islet were all returned to the base image and the positive area in each islet was recorded as a percentage of total islet size. For every pair of hormone stains (56 pairs), the Java JTS topography suite was used to calculate the intersection and the union of the stained regions. These areas were subsequently used to calculate Jaccard indices and relative areas of double-positive regions per islet (note that MICSSS is not the technology of choice for identification of bi- or tri-hormonal endocrine cells, *e.g.*, individual cells within pancreatic islets that co-express INS, GCG and/or SST; such analyses are better served by use of thinner tissue sections, concurrent immunofluorescent rather than successive brightfield staining, confocal microscopy, robust cell segmentation and additional visualization of endocrine cell lineage-specific transcription factors. Similar considerations also pertain to analyses of beta cell de-differentiation about which the current work has to remain ignorant). Additionally, we merged all hormone stains to find the total endocrine area per islet. Lastly, islet and endocrine cell type mass was calculated by multiplying relative fractions of islet areas or specific hormone staining areas with regional pancreas weights (PT or PH) (Supplementary Data 1).

**Immune cells.** Islet boundaries were expanded by 20 μm to mark the peri-islet region, using a watershed algorithm to ensure that the peri-islet boundaries of neighboring islets did not overlap. Within these regions, nuclei were detected *via* the QuPath implementation of Stardist[55], with 1 μm expansion for the cytoplasm. An object classifier was trained to detect CD45$^+$ immune cells. As the last stain in the MICSSS series, the CD45 image had the highest diffuse background staining within islets which furthermore varied between samples. To resolve this issue, the classifier used measurements of CD45 staining intensity in the nuclear, cytoplasmic, and membrane compartments, as well as the smoothed average of cells within a 100 μm neighborhood, and the difference between the cell's intensity and that of a 10 μm circular tile, including extracellular regions. After classification, the number and frequency of CD45$^+$ cells inside the islet and peri-islet region were calculated.

**CytoMAP analysis.** Islet measurements were exported to a .csv file and used for dimensionality reduction and clustering in the MATLAB implementation of CytoMAP[99]. The data input into UMAP[77] for dimensionality reduction included: the area of eight hormone stains, the total (union) endocrine stain area, islet-associated CD45$^+$ cell count, circularity, mean Delaunay area, and islet area. The total area, Delaunay area, and raw hormone areas (μm²) appeared log-normally distributed while the hormone staining areas as a percentage of islet areas were far from a normal distribution. Therefore, we used the log-transformed raw areas consistently. To balance the varying inputs, we calculated the z-scores for all of them. Pre-processing was performed in MATLAB. UMAP was run with the following parameters: n_neighbors = 50, min_dist = 0.1, n_epochs = 1000. Two rounds of DBScan clustering were performed on the UMAP output. First, to distinguish larger clusters, we used a high epsilon value (0.5) with a minimum number of 50 points. This yielded six clusters that largely corresponded to the visible clustering of the UMAP scatter plot (clusters 5 and 6 are in close proximity and were named V-A and V-BC). Second, to distinguish subclusters that were largely delineated by SST and GCG, we used a low epsilon (0.2) and a low minimum number of points (20). This yielded 16 clusters with some unclustered points. We used these

results to divide the six major clusters (I, II, III, IV, V-A, V-BC) into subclusters for deeper analysis. The islet cluster assignment was exported from CytoMAP into a .csv file and reuploaded into QuPath as a detection measurement using custom scripts. The cluster ID was converted into a class and then class-specific Delaunay clustering with a maximum search radius of 8 mm was performed.

**Spatial analysis.** For each slide, the automatically created tissue outline was manually edited to remove peripheral areas of adipose and connective tissue and fill in regions of missing tissue to determine the pancreas parenchymal area. This was exported from QuPath as a .geojson file and then imported into MATLAB along with the islet centroid locations. To calculate spatial distribution patterns in each slide, we used a modified Ripley's K function[100,101] that incorporates the tissue boundary and the total pancreas density. Circles are placed at each islet with varying radii (400–10,000 μm); looping through each islet, the number of other islets within that circle is counted; and the fraction of the circle area that lies inside the tissue boundary is used as a weighting factor for the number of points found. The adjusted number of points found at each distance is averaged across all islets in a slide. For Ripley analysis, this count is normalized to the number of points expected if islets were randomly distributed in the pancreas area - the average density multiplied by the area of the circle. If, at a given radius, the islets had on average the expected number of neighboring islets, the modified K value would be 1; however, if there were twice as many islets within a radius as expected, the modified K value would be 2. This allows for a comparison of spatial islet density enrichments even if the pancreatic tissue sections have different sizes, islet densities, and complexities. These data were further used for fractal spatial analysis following the method of Jo et al.[91] For each slide, search radii and boundary-adjusted islet counts were plotted on a log-log graph and the slope of the best-fit line was calculated, excluding radii ≤600 μm; the slopes, or fractal dimensions, of each slide were compared to demonstrate average changes in islet distribution with disease progression.

**Data visualization.** To ease data visualization and figure creation, the large matrix of islet measurements, including single and double hormone areas, locations, shape descriptors, spatial measurements and UMAP clusters were converted to .fcs files for visualization in FlowJo 10.10.0 (BDBiosciences) using the writeFCS function[102]. In some cases, contrast and brightness were adjusted for entire brightfield images using Adobe Photoshop (Adobe), and scale bars were added in Fiji/imageJ. Pseudofluorescence images were generated in QuPath using the spectrally separated AMEC channels per stain and the previously calculated affine transforms. Adobe Illustrator (Adobe) was used to prepare figures.

**HLA haplotype T1D risk classification**
HLA class I and class II haplotype data were classified for T1D risk according to published data and are summarized in Supplementary Data 1. HLA class II DR-DQ genotypes were binned based on T1D risk, high, moderate, neutral or protective, determined from individuals of European ancestry[59]. Bins were assigned numerical values for the purposes of this study, high risk (+2), moderate risk (+1), neutral (0), or protective (-1). Alternatively, HLA class II risk was classified using published odds ratios (OR) calculated for individuals of European (DR-DQ haplotype combinations[60]), African (individual DR-DQ haplotypes[61,62]), or admixed Hispanic/Latino descent (individual DR-DQ haplotypes[62]). HLA class I risk was classified for HLA-A and -B genotypes using odds ratios (OR) calculated from individuals of European descent[103].

**Quantification and statistical analysis**
Data analysis and graphical representation were performed in GraphPad Prism 9 and 10 (GraphPad Software). Normal distribution was

determined by D'Agostino-Pearson test. Statistical significance was assessed by unpaired or paired Student's t-tests or non-parametric Mann-Whitney U test as applicable for comparison between two groups; by one-way ANOVA with Tukey's multiple comparisons post hoc testing for analysis of more than two groups with normally distributed values; or by one sample t test using a hypothetical mean as indicated. Analyses of islet sizes were conducted using mixed effect models with subjects as random effects specifying a compound-symmetry correlation matrix in the model, and orthogonal contrasts were used for pair-wise comparison of the four donor groups. Summary data are displayed as scatter plots with mean, violin plots with median and quartiles, bar or line diagrams (mean±SE), frequency distributions (mean±95% CI error band), or correlation plots adopting the following convention: *$p < 0.05$, **$p < 0.01$, ***$p < 0.001$, ****$p < 0.0001$; ns or no symbol, non-significant.

Missing and excluded data: for the purpose of the present study, pancreatic islets are defined as endocrine objects ≥1,000 μm (~36 μm diameter); accordingly, we excluded smaller endocrine structures and single cells from all of our analyses. Information about HLA haplotypes and HbA1c values was available for most but not all donors (Supplementary Data 1 and Fig. S2a/c). Subcluster IV-E (Fig. 3a) is sample-biased and was not further considered. In some analyses of islet size frequency distributions and UMAP cluster-stratified islets, not all donors have islets present in all islet size bins or clusters; for islet property analyses, we additionally excluded values if <3 islets or <2 donors were represented in a size bin (Figs. 2m–q and S4a–h) or UMAP cluster; the resultant numbers of donors represented in each cluster or disease stage is indicated by color-coded values on top of respective x-axes in Figs. 7i, j, S5 and S6a, f, g (black: all donors; gray: reduced number of donors; red: instances with 2 donors only warranting interpretative caution, or absent/excluded data). CHGA and INS staining of Ctrl 6162 PT tissue sections and INS staining of AAb 6450 PT sections was notably weak; in the absence of CHGA and INS staining irregularities in the corresponding PH sections as well as normal ProINS and IAPP staining in both PT and PH sections, we attribute this observation to a technical staining issue and therefore excluded the respective PT CHGA and INS data from Figs. 1k, 2p, q, 3d, S1d, S2i–k, S3a, b, f, S4b, d and S5b, f, j. For calculation of endocrine cell type mass, the two 5-year-old donors Ctrl 6382 and T1DS 6209 were excluded in Figs. 1j and S2k, and the PH data from T1DS 6380 was not included in modified Ripley's K analyses due to very scarce islets (Figs. 7c and S8a).

**Reporting summary**
Further information on research design is available in the Nature Portfolio Reporting Summary linked to this article.

## Data availability
The supplemental information provided for this study includes detailed pancreas specimen information and donor metadata (Supplementary Data 1); properties of individual donor tissue sections and all ~25,000 islets captured therein and stratified according to pancreas region, donor group and UMAP sub/cluster affiliation (Supplementary Data 2); details for antibodies and MICSSS staining conditions (Table S1); and source data for all main and supplemental figures (source data file). Raw whole-slide brightfield images of pancreatic tissue sections captured at 40× will be provided by the lead contact upon request. All other data are available in the article and its Supplementary files or from the corresponding author upon request. Source data are provided with this paper.

## Code availability
Code used for image analysis using QuPath and MATLAB is accessible on GitHub: https://github.com/saramcardle/MICSSSPancreas or as a DOI on Zenodo: https://doi.org/10.5281/zenodo.17655136[104].

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

## Acknowledgements

We thank Dr. R. Brody (ISMMS Biorepository and Pathology CoRE) for provision of additional pancreatic tissue sections used for MICSSS optimization, Dr. O. Madsen (Novo Nordisk) for the gift of ProINS antibody, Dr. P. Bankhead (University of Edinburgh) for advice about QuPath customization, Dr. E. Bagiella (ISMMS Center for Biostatistics) for assistance with statistical analyses, Dr. B. Rosenberg (ISMMS Department of Microbiology) for advice about UMAP clustering, J. Gregory (ISMMS) for design of the model figure panel, and Dr. S. Richardson for detailed feedback on the manuscript. This research was supported by JDRF Fellowship 3-PDF-2018-575-A-N and a supplement (V.v.d.H.); Chan Zuckerberg Initiative grant DAF2019-198153 (S.M.); NIH grants CA224319, DK124165, CA234212, and CA196521 (S.G.); and NIH grants R01AI134971, R01DK130425, R21ES027916 and P30DK020541 (D.H.). To provide context for the multiplicity of observations reported here, we repeatedly cite consensus opinions summarized in authoritative reviews; we apologize to the authors whose pertinent primary contributions are not explicitly mentioned here. Most importantly, we thank the families of the organ donors for the gift of tissues.

## Author contributions

Conceptualization: V.v.d.H. and D.H.; investigation: V.v.d.H.; formal analysis: V.v.d.H., S.M., Z.M., K.C., and D.H.; software: S.M. and M.N.; writing - original draft: D.H.; writing - review and editing: all authors; visualization: V.v.d.H., S.M., and D.H.; resources: S.G., Z.M., A.L.P., I.K., and M.A.A.; funding acquisition: V.v.d.H. and D.H.; supervision: D.H.

## Competing interests
