## [Transparent Peer Review file · Nature Communications]

Integrated histopathology of the human pancreas throughout stages of type 1 diabetes progression

Corresponding Author: Professor Dirk Homann

Version 0:

Reviewer comments:

Reviewer #1

(Remarks to the Author)

in this manuscript the Authors performed a detailed, integrated histopathological assessment of the pancreas in 7 control subjects, 6 people without T1D but with autoantibody positivity, 8 with short duration T1D (< 2yrs) and 4 with longer duration T1D (8-11 yrs). They used sophisticated tools for data generation and analysis (including multiplexed brightfield immunohistochemistry, high magnification imaging, digital pathology, a semi-automated analysis pipeline of open-source pathology platform QuPath, integration thereof). They confirmed findings previously reported in T1D histopathology and described several novel findings, leading to the proposal of a revised natural history of T1D.

The study is of relevance and well written. Several points deserve clarifications:

1. The use of pancreatic head samples is a confounding factor, since apparently they contain, in non-precised but certainly variable proportions, parts derived from the ventral bud (uncinate process and posterior portion of the head) and parts derived from the dorsal bud (anterior portion of the head). This makes very difficult to clearly interpret the data generated and compare with the tail situation. My suggestion, if the "ventral" and "dorsal" head results cannot be distinguished, is to clear the head set of data, to have a more straight manuscript.
2. The lack of information on smaller islets (< 1,000 microm²) is indeed a major limitation, of which the Authors are well aware. This is particularly true since cluster II islets (small and mostly containing beta cells only) are key in the architecture of the study. The Authors are encouraged to provide at least some information on this, and hopefully not to let us wait for ongoing investigations
3. The population studied comprises different ethnicities, which the Authors mentioned without elaborating much; would it be possible to have more information on this?
4. The reviewer may be wrong, but does not see if there were cells containing both glucagon and insulin, and others with even triple positivities (as reported previously by other Authors); could this be clarified?
5. Previous work has shown that beta cell de-differentiation might occur before and after the onset of T1D (doi: 10.1016/j.cmet.2020.03.002; doi: 10.3389/fendo.2024.1427723); did the Authors check for this?

(Remarks on code availability)

I gave a look at the code, it provides the README file and an appropriate indication of the files; several of them are complex, and I could not make my mind fully.

Reviewer #2

(Remarks to the Author)

The authors assessed the characteristics of pancreatic tissue (head and tail regions) in non-diabetic, autoantibody positive (AAb), and type 1 diabetic (T1D) donors, these latter with short duration (T1DS) or long duration (T1DL) of the disease. Sections were sequentially stained for endocrine cell markers and for the immune cell marker CD45 and were analyzed using a semi-automated digital analysis pipeline. In addition, combinatorial patterns, characterizing the islet heterogeneity in the conditions studied, was assessed using a dimensionality reduction tool. This accurate approach together with the uniqueness of the samples obtained from the nPOD consortium, represent key strengths of the study in which the authors present a redefined model of the natural history of T1D histopathology. In particular, the progression from the non-diabetic to the T1D condition is characterized by changes in islet cell composition, immune cell infiltration and presence of insulinitis. Data support the concept that the modulation of ProINS and IAPP expression relates to pathogenetic processes in T1D and the kinetic of the disease is islet size dependent. In addition, the study and comparison of different regions of the pancreas

consolidates the concept of a multifocal and synchronized origin and progression of T1D. These novel aspects are of significance in the field since they may prompt towards investigations seeking to elucidate the pathogenesis of T1D. Furthermore, some data are confirmatory of previous studies that are acknowledged. The study is primarily technically oriented; however, it provides a detailed description of the workflow, from sample preparation to histological analysis and spatial analysis. Data interpretation and conclusions are well sustained by the overall analysis. The study is of interest, though some parts need refinement for clarity and smoothness.

- 1) The amount of data generated—both main and supplementary—is substantial and is discussed thoroughly, which in some cases makes it difficult to understand the overall message. We would suggest refining certain sections where possible, relocating potentially redundant information to the supplementary materials. For example, in Figure 1 panels B, G, and I could be simplified by including only one representative image, while more detailed illustrations of the computational analysis could be moved to the supplementary materials. Panel C may be removed, and the staining order could be described in the figure legend. If possible, please revise all the Figures accordingly.
- 2) In Results, the first two paragraphs (page 4, lines 72-112) are descriptive of the methodology, which the authors propose as a novel analytical approach. However, they might be moved to the Methods section, as they are not descriptive of the results. In addition, the statement “Pancreatic islets, defined here as endocrine cell clusters $\geq 1,000 \mu\text{m}^2$ (~10 cells/~36 μm diameter)” (line 90) can be omitted since included in Methods (page 34, line 1108).
- 3) In Discussion, the statement “Based on the proposition that islet size distribution can serve as a simple organizing principle for the substantial heterogeneity of islet composition, our observation that alterations of islet architectural features in early T1D are independent of islet size provides initial support for this notion” (page 16, line 497) is not clear, could the authors explain better?
- 4) In Methods (page 31, line 1004), it is stated “note that there is no official consensus about a minimal islet size that distinguishes it from smaller endocrine cell clusters; rather, our choice of $1,000\mu\text{m}^2$ as a suitable threshold reflects a compromise based on historical conventions”. Could the authors add supporting reference/s?
- 5) The number of islets studied is impressive (about 500 islets per tissue section, about 25,000 in total). However, the number is not consistent among the cohorts (Fig.7, Fig.S8). These reviewers suggest providing a simple table in which is reported the number of islets studied in each section (H and T) for each donor, together with the area of the respective pancreas/acinar section.

(Remarks on code availability)

Reviewer #3

(Remarks to the Author)

Summary

Heide et al perform extensive histopathological and computational analysis of the pancreata of non-diabetic, autoantibody positive (AAb) and T1D organ donors. Their analyses focused on islet hormonal composition, the geometric architecture of islets, patterns of CD45+ cell infiltration, and the spatial distribution of islets throughout the pancreas. Major findings include 1) changes to islet size and geometry during T1D progression that are evident even in islets that are not visibly infiltrated by immune cells; 2) changes in islets as early as stage I/II that develop further during T1D. 3) clustering of islets (perhaps of the same “type”) in pancreatic neighborhoods accompanied by tissue-wide islet depletion. Together, these data further characterize the progression of T1D pathology, and the computational methods may have practical value for the analysis of large pancreas imaging datasets.

Major Strengths

1. The focus on geometric shape descriptors is interesting. Several observations regarding their changes over the course of T1D are compelling.
2. Generation of a comprehensive dataset that will be a valuable resource for the T1D community.
3. An emphasis on the analysis of AAb cases which are very important but understudied.

Major weaknesses

4. Several key claims could be better supported (see numbered sections after 12. for more details):
5. Key findings are limited. Specifically, Figure 1 merely describes the method but contains no findings. The first half of Figure 2 is primarily validating the technique by recapitulating known changes. Figure 3 introduces sub-clusters of insulin-containing islets (ICIs) and insulin-deficient islets (IDIs) but does not report any meaningful differences between them. In Figure 4, the same data is plotted in A-D just organized differently. In Figure 5, it's not clear that breaking down the analysis by sub-cluster is of value; it is not truly a finding that immune cells are enriched in specific islet clusters because immune cells were used to define the islet clusters. Figures 6-7 are interesting, but they should be interpreted conservatively.
6. In the text, I found many of the conclusions very challenging to understand due to the writing.

Minor weaknesses

7. Some of the findings have been previously published but are not cited or if cited, the specific findings are not engaged with:
 - a. Geometric measurements:
 - i. Seiron et al, measure islet diameter, circularity and islets/mm.
 - ii. Damond et al, 10.1016/j.cmet.2018.11.014 examine islet solidity and extent over different stages of disease. They also show reductions in both in T1D.
 - b. ProINS/IAPP reduction in AAb
 - i. Teresa Rodriguez-Calvo et al, <https://doi.org/10.2337/db16-1343> show that proinsulin is increased in AAb+ vs Ctl cases, in

contrast to the your data. Potential explanations for this discrepancy should be discussed.

c. Immune infiltration in AAb cases

i. Insulinitis is present in a small fraction of early onset T1D cases and an even smaller fraction of autoantibody-positive cases. In't Veld et al, <https://doi.org/10.2337/db07-0416>.

d. Reduction in islet areas with T1D

i. Wilcox et al, 2010, 10.1007/s00125-010-1817-6

e. IDIs with immune cells

i. Barlow et al, ELife, <https://doi.org/10.7554/eLife.100535.3>

f. Clustering of islets regionally

i. Barlow et al, ELife

8. Lack of explanation or even speculation of the biological processes underlying changes in geometric shape descriptors. It would be valuable to show example images of islets and have a pathologist to review them.

9. I am skeptical that this method will be useful in practice given the significant effort required to perform the staining and image registration. There are now many multiplexed immunofluorescence platforms that seem more feasible.

10. It would be very valuable to demonstrate that the geometric changes are not artifacts of the image processing pipeline. This could be done using published highly multiplexed datasets from Damond (Cell Metabolism) et al and Barlow et al (ELife).

11. The Discussion section should cite relevant figures with each key claim to make it easier for the reader to evaluate the support for the statement.

Support for key claims continued

Major claims (broken down point by point from Abstract/Discussion):

12. Line 64-68 are very general and conceptual but not clear. They could be more specific and centered on key findings.

13. "new histopathological correlates for the stage 1/2 pancreas suggest AAb subjects are "on cusp" of developing T1D hallmarks":

a. "higher frequencies of islets with ≥ 1 associated immune cell". Immune infiltration in AAb cases has been published but it is even rarer than in early-onset T1D so this is hardly a feature of AAb subjects or islets.

b. "early targeting of small, including "GCG-deficient" islets". The claim that GCG-deficient islets are targeted early is circumstantial and incomplete (see section 19).

c. "collapse of islet cluster I magnitude". The magnitude of Cluster 1 vs 2 is likely driven by CD45 count so this is the same point as a) above.

14. histopathology of T1D:

a. "tissue-wide distributed rather than strictly localized disease process". It is unclear what supports this. Is this a reference to the similarities across PT and PH?

b. "alterations of islet architectural features in early T1D are independent of islet size" This argues against the claim that that islet size distribution is important. It is not clear what a 'simple organizing principle' means practically.

c. "absence of a direct correlation between islet composition and associated immune cell abundance". The fact that small islets without CD45+ cells also decrease ProINS expression could be easily explained since only 2D sections are analyzed and immune infiltrate is typically sparse and concentrated at one point. It is reasonable to expect a "dropout effect" for small islets.

d. "consider the entire spectrum of immune cell associations with islets beyond those affected by insulinitis". Unclear

e. Lines 504-508. Unclear

Major claims (broken down from Results):

Unclear:

15. Line 178: "absence of alpha cells...suggesting that "GCG-deficient" islets constitute an integral part endocrine pancreas physiology". If the cited work shows no effect of alpha cells on beta cells, how can they conclude GCG-deficient islets are integral.

16. The observation in Figure 6 that the distribution of islets can change from being uniform to aggregated is quite interesting but in the exact text in line 428-431 "Since islet locations are immutable... cumulative areas, and mass", I don't understand what the specific claim is because the definitions of "tissue regions with unperturbed islet densities" or "islets at shorter distances" is unclear. So, I can't confidently evaluate the merit of the claim.

Overinterpreted/unsubstantiated

17. line 150 "they emphasize a profound alteration of the endocrine pancreas in the T1DL stage beyond the loss of beta cells". It's not clear what is profound about the findings.

18. Line 181: Authors report a trend in ProINS^{hi} islets from Ctrl to AAb (Figure 2D). They would need to show side by side the %ProINS/GCG-negative and %ProINS/GCG-positive islets to say the GCG-deficient are more vulnerable than GCG-positive islets.

19. Line 252: Are Ins⁻ and Ins⁺ islets included in 2Q (ProINS through SST columns)? Without knowing, it's hard to separate the fraction of beta cells in each islet vs the fraction of islets that are completely insulin deficient.

a. "reflects an early loss of beta cells especially in smaller islets". By "early loss", do you mean in Ctrl vs AAb or Ctrl vs T1DS?

b. "preservation of residual beta cell mass in larger T1DS islets". The idea that larger islets preserve beta cell mass is interesting, but the order of events can't be discerned from these data. It is possible that islets shrink after losing beta cells or they all shrink in parallel as disease progresses, not necessarily that smaller islets are targeted first. It would be good to invoke the insulinitis data to make the case that islet size effects targeting.

20. Line 304: "early targeting of smaller ICIs and later appearance of larger IDIs" First, it's unclear what is meant by "appearance". Second "larger" is confusing because IDIs are smaller than ICIs. If I understand correctly, the claim is that the gradual increase in islet size (in ICIs and IDIs) over the course of T1D suggests that smaller islets are losing insulin and becoming IDIs before larger islets do. Again, I think this is interesting but should not be overstated. If this were the only cause, one would expect IDIs at late stage T1D to be similar size to ICIs (once all islets lose insulin). To be fair, the authors do not claim that size-dependent disease kinetics is the only cause. Conclusions should be qualified and elaborated upon in the discussion.

21. Line 340-342: 5A looks like there are 3 AAb cases with insulinitis and these are the three that also have CD45+ cells. Text should be clarified to explain that CD45+ cells can be found in specific AAb cases, not that CD45+ cells are elevated at the stage generally.

22. Line 388: Is there evidence in this dataset that CD45+ cells are recruited to islets directly? For example, are there increases in CD45+ cells in acinar tissue outside islets? If not, it could help substantiate that immune cells are recruited actively at the AAb stage.

23. Line 443. "Collectively, our observations support the notion that the typically regionalized patterns of T1D represent residuals of a tissue-wide islet depletion, the extent of which is partially obscured by its fundamentally dispersed nature." Can volumetric data be used to validate this? Is there any evidence in literature or biological mechanism for islets "disappearing" vs growth of exocrine cells?

Methodology

1. The islet and cell mass calculations are unreliable. IHC is not quantitative enough to evaluate the extremely large dynamic range of hormone secretion. It is better to simply show the pancreatic mass and islet/endocrine cell area separately.

2. It is unclear the extent to which the differences in percentages between PH and PT throughout the manuscript are due to the PPY islets in the uncinate process. Would it not be clearer to characterize the uncinate process and then exclude this area from all following analyses?

3. Panels S3F using Jaccard index (Line 189) are not meaningfully mathematically different from the fraction of those hormones as reported in the main figure.

4. Figure 2K left vs right. How can AAb have a different islet distribution than the other 3 groups on the left but T1DL has the different distribution than the other 3 in the right?

5. To ensure that differences in sphericity and circularity that accompany T1D are not an artifact of the image processing, (e.g perhaps the reduced hormone expression and imprecise thresholding leads to cell drop out and hence inaccurate measurements of the perimeter and other features). It would be helpful to show multiple images of islets that represent different sphericities, aspect ratios, and solidity. A quantitative validation would also be helpful. For example, since chromogranin is unchanged, you could calculate the geometric shape descriptors only using chromogranin.

6. 2Q should contain error bars, ideally via bootstrapping.

7. In Figure 2D, is the z-score computed column wise or row wise, and separately or pooled between PT and PH islets.

8. Is figure 4A intended to show the frequency of islets in each cluster or the ProINS expression over time. It doesn't serve either purpose well in its current form.

9. Are only hormones used in islet UMAP or geometric features as well?

10. In Figures 3-4, It's essential to show the features that were used for the umap and their means in each cluster. The umap uses the islet area and CD45+ cell count which clearly will separate clusters 1 and clusters 2 but this was not clear at all from the text or figures.

11. Figure 5E needs statistics

12. In Figure 7, the use of Delaunay triangulation and 'isolated islets' does not effectively describe the regionalization of islet clusters because it is highly influenced by the abundance of the clusters. A manual approach could include repeatedly sampling neighboring islets vs random islets from the same tissue and compare the enrichment of a given cluster in the sample. Alternatively, a spatial correlation method such as a cross-K function would be more direct and appropriate.

Other comments:

Is any prescreening performed on the tissues for insulinitis before selecting blocks for the study? If so, this minimizes the ability to use CD45+ cells as a diagnostic as proposed in line 387.

Do islet cluster foci in Figure 7 comport to lobules boundaries?

To validate the utility of geometric shape descriptors, can they be measured in HandE images and applied to large datasets?

(Remarks on code availability)

Reviewer #4

(Remarks to the Author)

I co-reviewed this manuscript with one of the reviewers who provided the listed reports. This is part of the Nature Communications initiative to facilitate training in peer review and to provide appropriate recognition for Early Career

Researchers who co-review manuscripts.

(Remarks on code availability)

I am not able to assess the code, as it falls outside my technical expertise and background. Therefore, I cannot evaluate the reproducibility of the results or the usability of the code for the broader community.

Version 1:

Reviewer comments:

Reviewer #1

(Remarks to the Author)

The Authors have satisfactorily addressed all the points raised by this reviewer. They are nevertheless invited to succinctly mention the issues discussed in their responses to point 4 (co-stainings) and point 5 (beta cell dedifferentiation) in the manuscript, for the sake of completeness.

(Remarks on code availability)

The code can be now better appreciated.

Reviewer #2

(Remarks to the Author)

The manuscript has been revised. Figures have been modified in line with reviewers' recommendations, and a supplementary table has been included to present the quantitative data underlying Fig. S2e, Fig. S2g, and Figs. 1i and S2h (including total tissue area, parenchymal tissue area, and the number of islets analyzed per section). The issues raised in comments 3 and 4 have been comprehensively addressed, and the response to comment 2 is convincing.

(Remarks on code availability)

Reviewer #4

(Remarks to the Author)

(Remarks on code availability)

Response to Referees

We wish to thank the Reviewers for their time and effort to assess our manuscript, and we appreciate the opportunity to respond to their comments. Below, please find our point-by-point reply in which **all callouts to manuscript lines, figures and tables refer to the revised manuscript**, unless noted otherwise (Reviewer comments are in black font, our replies are in blue font). In the revised manuscript itself, all changes are highlighted by gray background shading to readily identify the edits made throughout the entire text, and changes to accompanying figures are summarized in our reply below.

Reviewer #1 (Remarks to the Author)

in this manuscript the Authors performed a detailed, integrated histopathological assessment of the pancreas in 7 control subjects, 6 people without T1D but with autoantibody positivity, 8 with short duration T1D (< 2yrs) and 4 with longer duration T1D (8-11 yrs). They used sophisticated tools for data generation and analysis (including multiplexed brightfield immunohistochemistry, high magnification imaging, digital pathology, a semi-automated analysis pipeline of open-source pathology platform QuPath, integration thereof). They confirmed findings previously reported in T1D histopathology and described several novel findings, leading to the proposal of a revised natural history of T1D.

The study is of relevance and well written. We thank the Reviewer for this assessment.

Several points deserve clarifications:

The use of pancreatic head samples is a confounding factor, since apparently they contain, in non-precised but certainly variable proportions, parts derived from the ventral bud (uncinate process and posterior portion of the head) and parts derived from the dorsal bud (anterior portion of the head). This makes very difficult to clearly interpret the data generated and compare with the tail situation. My suggestion, if the "ventral" and "dorsal" head results cannot be distinguished, is to clear the head set of data, to have a more straight manuscript. [The Reviewer is correct to assert that the pancreatic head \(PH\) sections used in our study were not selected according to a defined preponderance of uncinata process tissue therein; accordingly, some of our experimental readouts exhibit greater variability \(e.g., proportion of PPY-containing islets\). Although the "PPY-rich" PH lobe has been recognized for almost half a century \(and some of the older histological analyses in fact explicitly distinguish between posterior and anterior PH sections\), more recent reports in the context of T1D histopathology tend to emphasize differences or similarities between PH and pancreas tail \(PT\) \(and/or pancreatic body\) in the absence of PPY stains and thus typically without any reference to the uncinata process. We have structured the data presentation in our manuscript starting with more traditional outcome metrics \(e.g., cumulative staining areas of individual hormones\) and then proceeding to more contemporary data display as afforded by our multiplex staining \(e.g., "single-islet" UMAPs\). Accordingly, we first confirm and refine known PH vs. PT differences using the former analysis modalities, and subsequently demonstrate that use of the latter modalities \(i.e., UMAP cluster stratification and their "back-projection" onto pancreatic tissue sections; cluster III/IV islet "back-projection" specifically delineates uncinata process proportions in all 25 PH sections as shown in **Figs.7h & Sh8**\) suggests "an essentially identical endocrine organization across pancreas regions that locates residual differences specifically to the uncinata process in the PH" \(lines 502-504; also note that our main figures foreground PT analyses with a good portion of the PH data relegated to the Supplement\). Importantly, and in response to the Reviewer's concern, we have now included a new supplementary **Fig.S5i/j** that provides a comprehensive summary of the main data in support of our contention. We believe this conclusion to be of considerable relevance for an inclusive assessment of pancreatic histopathology throughout the course of T1D development, and we therefore prefer to maintain the respective data and their discussion in the manuscript. Perhaps our report will also incentivize future efforts to experimentally distinguish anterior and posterior PH portions, much as has been done in the older literature using more limited analytical tools.](#)

2. The lack of information on smaller islets (< 1,000 microm²) is indeed a major limitation, of which the Authors are well aware. This is particularly true since cluster II islets (small and mostly containing beta cells only) are key in the architecture of the study. The Authors are encouraged to provide at least some information on this, and hopefully not to let us wait for ongoing investigations. Questions about our choice of a 1,000µm² islet size threshold have also been raised by the other Reviewers, and we contend that this matter is in fact more complicated than at first may appear. We will therefore provide a more detailed account covering this topic.

As we note in lines 1,056-1,058 of the Methods section, “*there is no official consensus about a minimal islet size that distinguishes it from smaller endocrine cell clusters; rather, our choice of 1,000 μm^2 as a suitable threshold reflects a compromise based on historical context*”. Islet size ranges have historically been determined using a variety of methods including traditional two-dimensional (2D) histology, three-dimensional (3D) tissue analyses and examination of isolated islets providing a range of estimates for islet diameters, areas and volumes, as well as diameter calculations based on the assumption of circularity or sphericity = 1 (*i.e.*, diameters are calculated for circle- or sphere-equivalent islet areas or volumes). For the purpose of providing the following brief synopsis, we feature all values as area measurements (*i.e.*, in μm^2).

For some 40 years now, we have known that “*in spite of the distinction previously made between islets of Langerhans and so-called extra-islet cells, they are in fact both part of a continuous distribution*” (PMID: 3526638). Nevertheless, the nominal distinction of “small islets”, “extra-islet cells” or “endocrine cell clusters” has persisted, thresholds appear to vary according to investigator preference, and some authors seemingly suggest that even single endocrine cells can be considered “islets” (“*Islet sizes are in a wide range from a single endocrine cell to a large islet consisting of several thousand cells*”, PMID: 22653677). Proposals for an applied threshold for islet capture range from 2,000 μm^2 (PMID: 20606719, PMID: 39720415) to 1,275 μm^2 (PMID: 20185817, PMID: 30936150; the latter publication explicitly stating that “*while singly scattered endocrine cells and small clusters are always included, in this study, only islets >40 μm in diameter [i.e., >1,257 μm^2] were examined for meaningful sizes of islets*”) to 1,000 μm^2 (PMID: 17708340, PMID: 21641386), 962 μm^2 (PMID: 40705030), and further down to 707 μm^2 (PMID: 40502779), 660 μm^2 (PMID: 38632302), and 490 μm^2 (PMID: 31493350).

A 2018 review of both 2D and 3D islet size quantifications by Huang *et al.* indicates that islets are found in the range of 707 μm^2 to >125,660 μm^2 (PMID: 29954219), and Ionescu-Tirgoviste *et al.* (PMID: 26417671) provide an in our view sensible islet size allocation into four major bins (<1,000 μm^2 , 1,000-10,000 μm^2 , 10,000-100,000 μm^2 , and >100,000 μm^2). While the field would certainly benefit from a consensus and more consistent use of terminology, the above outline suggests that in the absence thereof, our choice of a 1,000 μm^2 for the present study represents a reasonable compromise (as requested by Reviewer #2, we added a selection of the above cited references in line 1,058 to provide historical context). Please see our replies to individual Reviewer comments below for additional detail.

In regard to Reviewer #2’s comments about “smaller islets”, we note that “small endocrine objects”, long recognized and equally long neglected, have only this year become a topic of serious interest in the context of T1D (PMID: 38922355, PMID: 40705030 and <https://www.biorxiv.org/content/10.1101/2025.04.11.648319v1>). In fact, the latter two manuscripts examined a large number of archival images of pancreatic tissue sections stained for insulin (INS) and glucagon (GCG) and taken together, came to the same conclusion: “small endocrine objects”, especially those expressing INS in the absence of GCG, constitute important targets in the early stages of T1D pathogenesis. This observation aligns beautifully with our own findings that the smallest islets (*i.e.*, those just above the size threshold of 1,000 μm^2) are consigned to the same fate, *i.e.*, they are the targets whose beta cell content is preferentially reduced and even depleted already in the autoantibody* T1D stage 1/2 and beyond. These findings also support our contention that the novel histopathological correlates we describe at early disease stages are consistent with a chronic-progressive disease course in which the autoimmune destruction of beta cells proceeds at a relatively constant rate throughout all stages of T1D (as per a very recent hypothesis advanced in PMID: 39106185).

So, while we are in the process of conducting similar and additional “small endocrine object” analyses with our own data sets, the work is far from concluded (not least because we need to consider multiple additional metrics beyond INS and GCG stains) and furthermore would considerably expand on an already long and data-rich manuscript. And while we certainly agree with the Reviewer that additional “small endocrine object” analyses can contribute to a refined understanding of the T1D natural history, we prefer to conclude this work without haste and present new data in a follow-up manuscript with the adequate scope and depth that the topic deserves. We also emphasize that despite our exclusion of endocrine objects <1,000 μm^2 , we capture, count and characterize virtually all islets in all tissue sections from all donors above this threshold yielding a respectable total number of islets (~25,000) that, we believe, contributes to the robustness of our study design, its execution and the respective conclusions.

Lastly, please note that cluster II islet size medians are actually the largest in our UMAP cluster stratification. However, the Reviewer is correct in that we propose a particular vulnerability of small islets to pathogenic processes, *cf.*, **Figs.2c/g/h/q & 6f/g**; also, see our replies to respective comments by Reviewer #3 below. We

realize that at first blush, there appears to be contradiction in stating that small islets are a preferred early autoimmune target while the overall larger cluster II islets constitute “*the apparent histopathological locus of CD45⁺ cell infiltration*” (lines 402-403). However, our analyses resolve this seeming contradiction by expanding our histopathological focus beyond instances of insulinitis and considering a progressive reduction of beta cell hormone areas rather than only complete beta cell loss.

3. The population studied comprises different ethnicities, which the Authors mentioned without elaborating much; would it be possible to have more information on this? All available information about donor ethnicities is provided in Table S1 and is taken into account for our HLA risk score calculations (see Tables S1 & Fig.S2a); unfortunately, our overall group sizes are too small to allow for a meaningful donor subset analysis stratified according to race/ethnicity.

4. The reviewer may be wrong, but does not see if there were cells containing both glucagon and insulin, and others with even triple positivities (as reported previously by other Authors); could this be clarified? Bi- and tri-hormonal pancreatic endocrine cells constitute an intriguing field of study to which the present study, however, cannot convincingly contribute. Although our image overlay of the same MICSSS-stained tissue sections achieves cell-level accuracy, we decided to forgo an enumeration of islet endocrine cell types due to their tight spatial organization in the islets and the considerable challenges associated with robust cell segmentation; we therefore decided to report specific hormone staining areas instead. A convincing characterization of bi- and tri-hormonal endocrine cells requires, in our opinion, the use of targeted fluorescence-based multiplex staining panels (e.g., INS, GCG, SST, ideally in combination with lineage-specific transcription factors and robust cell segmentation) that do not rely on image overlays.

5. Previous work has shown that beta cell de-differentiation might occur before and after the onset of T1D (doi: 10.1016/j.cmet.2020.03.002; doi: 10.3389/fendo.2024.1427723); did the Authors check for this? This is an interesting question. However, we have not included stains for endocrine cell lineage-specific transcription factors or similar targets of interest and we therefore cannot make any informed speculations regarding potential beta cell de-differentiation in the course of T1D development and progression.

(Remarks on code availability) I gave a look at the code, it provides the README file and an appropriate indication of the files; several of them are complex, and I could not make my mind fully. We have updated the github-deposited README file to better explain the logical flow for each script (https://github.com/saramcardle/Image-Analysis-Scripts/tree/master/MICSSS_Pancreas).

Reviewer #2 (Remarks to the Author)

The authors assessed the characteristics of pancreatic tissue (head and tail regions) in non-diabetic, autoantibody positive (AAb), and type 1 diabetic (T1D) donors, these latter with short duration (T1DS) or long duration (T1DL) of the disease. Sections were sequentially stained for endocrine cell markers and for the immune cell marker CD45 and were analyzed using a semi-automated digital analysis pipeline. In addition, combinatorial patterns, characterizing the islet heterogeneity in the conditions studied, was assessed using a dimensionality reduction tool. This accurate approach together with the uniqueness of the samples obtained from the nPOD consortium, represent key strengths of the study in which the authors present a redefined model of the natural history of T1D histopathology. In particular, the progression from the non-diabetic to the T1D condition is characterized by changes in islet cell composition, immune cell infiltration and presence of insulinitis. Data support the concept that the modulation of ProINS and IAPP expression relates to pathogenetic processes in T1D and the kinetic of the disease is islet size dependent. In addition, the study and comparison of different regions of the pancreas consolidates the concept of a multifocal and synchronized origin and progression of T1D. These novel aspects are of significance in the field since they may prompt towards investigations seeking to elucidate the pathogenesis of T1D. Furthermore, some data are confirmatory of previous studies that are acknowledged. The study is primarily technically oriented; however, it provides a detailed description of the workflow, from sample preparation to histological analysis and spatial analysis. Data interpretation and conclusions are well sustained by the overall analysis. We thank the Reviewer for this assessment.

The study is of interest, though some parts need refinement for clarity and smoothness.

1) The amount of data generated—both main and supplementary—is substantial and is discussed thoroughly, which in some cases makes it difficult to understand the overall message. We would suggest refining certain

sections where possible, relocating potentially redundant information to the supplementary materials. For example, in Figure 1 panels B, G, and I could be simplified by including only one representative image, while more detailed illustrations of the computational analysis could be moved to the supplementary materials. Panel C may be removed, and the staining order could be described in the figure legend. If possible, please revise all the Figures accordingly. We appreciate these thoughts and in response to them, we have modified the figures as follows:

Fig.1: former panel B has been moved to the supplement (now Fig.S1a), and former panel I has been reduced from 8 to 4 islet images (now Fig.1h). While we considered including the information in former panel C (MICSSS staining order) in the figure legend, overall figure layout constraints provided us the opportunity to maintain it as Fig.1b. Importantly, and also in response to a concern by Reviewer #3, we have moved experimental data from the previous Fig.2 to what is now Fig.1i-k.

Fig.2: the figure has been reduced by removal of previous panels A-D, with additional changes made in response to Reviewer #3 discussed below.

Fig.4: we removed former panel C, now presented in the supplement as Fig.S6b (also, cf., comment by Reviewer #3 below).

Figs.5-6: to improve the data presentation of former Fig.5, we have created two new figures, i.e., Figs.5 & 6. Fig.5 now features data on insulinitis and insulitic islet properties and includes the new panels b/right (insulinitis frequencies of ICI and IDI subsets) addressing concerns raised by Reviewer #3. Fig.6 now displays aspects of UMAP cluster-stratified and overall islet-associated immune cell burden, with former Figs.6 & 7 relabeled accordingly.

Overall, our revised design of the main figures seeks to capture and summarize specific topics (i.e., Fig.1: methodology and basic analyses; Fig.2: more complex analyses based largely on traditional analysis modalities and/or simple islet subset stratification; Fig.3: introduction of “single-islet” UMAP clustering and description of pertinent cluster features; Fig.4: UMAP cluster magnitude “dynamics” throughout T1D progression; Fig.5: insulinitis and insulitic islet properties; Fig.6: UMAP cluster-stratified and overall islet associated immune cell burden; Fig.7: analyses of spatial islet distributions; and Fig.8: additional spatial analyses emphasizing immune cell:islet associations and a revised T1D natural history model). We hope this organization is sensible and have refrained from taking full advantage of all 10 display items permitted by *Nature Communications* policies.

2) In Results, the first two paragraphs (page 4, lines 72-112) are descriptive of the methodology, which the authors propose as a novel analytical approach. However, they might be moved to the Methods section, as they are not descriptive of the results. In addition, the statement “Pancreatic islets, defined here as endocrine cell clusters $\geq 1,000\mu\text{m}^2$ (~10 cells/~36 μm diameter)” (line 90) can be omitted since included in Methods (page 34, line 1108). The Reviewer is correct that lines 72-112 primarily discuss our experimental approach and analytical strategies. While a considerably more detailed description is provided in the Methods section, we believe that the succinct outline in the Results section is warranted to introduce the reader to the scope and focus of our subsequent data presentation. Emphasizing the nature, quality, extent, and limitations of our principal analytical readouts is in our view essential as it supports the robustness of each of our conclusions. Lastly, as detailed in item 1) above, we have condensed information regarding methodology and analysis strategies in Fig.1 and added experimental results in respective panels i-k.

Regarding the proposed threshold of \$1,000\mu\text{m}^2\$ for islet capture, please see our response to Reviewer #2 above the above. In light of the attention that this topic has generated in the review process, we prefer to briefly mention our choice of threshold for islet capture in the Results section for contextual clarity.

3) In Discussion, the statement “Based on the proposition that islet size distribution can serve as a simple organizing principle for the substantial heterogeneity of islet composition, our observation that alterations of islet architectural features in early T1D are independent of islet size provides initial support for this notion” (page 16, line 497) is not clear, could the authors explain better? The statement has been removed. Its content has been rephrased and is now featured in lines 501-502 and 513-515 (see also comment by Reviewer #3, section 14.b.).

4) In Methods (page 31, line 1004), it is stated “note that there is no official consensus about a minimal islet size that distinguishes it from smaller endocrine cell clusters; rather, our choice of $1,000\mu\text{m}^2$ as a suitable threshold reflects a compromise based on historical conventions”. Could the authors add supporting reference/s?

References have been provided as suggested in line 1,058. Further information about the topic of islet sizes and the use of varying size thresholds used for definition of islets is provided in our reply to Reviewer #2 above.

5) The number of islets studied is impressive (about 500 islets per tissue section, about 25,000 in total). However, the number is not consistent among the cohorts (Fig.7, Fig.S8). These reviewers suggest providing a simple table in which is reported the number of islets studied in each section (H and T) for each donor, together with the area of the respective pancreas/acinar section. The Reviewer is correct to note that there is islet count variability within donor groups as well as an overall reduction of islet numbers with T1D progression, the latter observation also expressed as “islet densities” and consistent with the cited literature. We have graphically displayed the parameters discussed by the Reviewer in Fig.S2g (absolute number of islets captured in each donor PT/PH tissue section), Fig.S2e (absolute size of parenchymal tissue section in mm², and fraction of parenchymal tissue area of total tissue section area in percent for each donor PT/PH sample), and Figs.1i & S2h (islet densities expressed as number of islets per mm² parenchymal pancreas area for each PT/PH tissue section). As suggested by the Reviewer, we have now added a new tab to Table S2 featuring these data in tabular format.

In the case of our insulinitis discussion, individual donor-stratified islet numbers captured by our analyses are also featured in the Fig.5a panels themselves. Importantly, please note that we did not select for or exclude any islets; rather, we captured and analyzed all islets in all PT/PH tissue sections from all donors (in regard to the number of tissue section displays featuring islets color-coded according to UMAP cluster affiliation, we show all 50 sections in Figs.7h & S8a; the same applies to all 50 sections displaying the islet-associated CD45 cell burden in Figs.8a & S8b).

Reviewer #3 (Remarks to the Author)

With all due respect, the Reviewer provides a very extensive itemized list of comments, critiques and concerns that we at times found somewhat difficult to address, in part due to an occasional lack of clarity and assertions regarding statements that we did not make. While we have sought to address each item individually, issues of repetition resulted in a need to use abundant cross-referencing of our arguments throughout this reply. We hope that our responses nevertheless help to clarify the issues raised by the Reviewer.

Summary

Heide et al perform extensive histopathological and computational analysis of the pancreata of non-diabetic, autoantibody positive (AAb) and T1D organ donors. Their analyses focused on islet hormonal composition, the geometric architecture of islets, patterns of CD45+ cell infiltration, and the spatial distribution of islets throughout the pancreas. Major findings include 1) changes to islet size and geometry during T1D progression that are evident even in islets that are not visibly infiltrated by immune cells: 2) changes in islets as early as stage I/II that develop further during T1D. 3) clustering of islets (perhaps of the same “type”) in pancreatic neighborhoods accompanied by tissue-wide islet depletion. Together, these data further characterize the progression of T1D pathology, and the computational methods may have practical value for the analysis of large pancreas imaging datasets.

Major Strengths

1. The focus on geometric shape descriptors is interesting. Several observations regarding their changes over the course of T1D are compelling.
2. Generation of a comprehensive dataset that will be a valuable resource for the T1D community.
3. An emphasis on the analysis of AAb cases which are very important but understudied.

We thank the Reviewer for this assessment.

Major weaknesses

4. Several key claims could be better supported (see numbered sections after 12. for more details): Please see our responses below as well as item 12. following.

5. Key findings are limited. Specifically, Figure 1 merely describes the method but contains no findings. The Reviewer is correct to note that **Fig.1** describes our experimental methodology and analytical strategies. Accurate demarcation of tissue section areas, capture of islets, derivation of shape descriptors, delineation of hormone staining areas and identification of CD45⁺ cells are essential for the generation of downstream readouts, and we argue that this information needs to be displayed prominently rather than relegated to the supplement. For further details, please see our response to Reviewer #2, item 1) where we detail our reduction of original **Fig.1** display items and the addition of experimental data in the new respective panels i-k.

The first half of Figure 2 is primarily validating the technique by recapitulating known changes. Again, the Reviewer is correct in noting that the first half of former **Fig.2** contains data confirming expected histopathological changes in T1D development and progression. However, a prominent display of the data we selected for what are now **Figs.1i-k & 2a** is warranted for three related reasons:

1. The principal goal of our study is to derive more detailed insights into the natural history of T1D by conducting comprehensive histopathological analyses. For this purpose we have selected a multiplexed IHC approach, acquisition of high-resolution whole-slide images and, importantly, we developed and applied a semi-automated image analysis pipeline since comprehensive manual evaluation of the large data sets is not possible. Therefore, we need to convincingly demonstrate that our strategy yields results that readily align with the major established hallmarks of T1D histopathology; only then can we be confident that the novel observations reported here are grounded in a robust automation of image analysis tasks (and we state as much in lines 231-234).
2. Despite shared histopathological hallmarks across various donor cohorts in different studies, unique aspects pertaining to the donor groups under investigation need to be considered to better appreciate both commonalities and differences in comparison to the published literature; thus, key data to this extent needs to be readily accessible and discernable.
3. The reporting of “standard outcomes” (*i.e.*, “*recapitulating known changes*”) also provides important context to better appreciate less frequently reported outcomes (*e.g.*, **Fig.1k** displays the progression of islet endocrine content not only for INS and GCG but for all major hormones across T1D stages).

Figure 3 introduces sub-clusters of insulin-containing islets (ICIs) and insulin-deficient islets (IDIs) but does not report any meaningful differences between them. We respectfully disagree. The majority of **Fig.3** (and the related **Fig.S4j**) is in fact devoted to detail UMAP cluster differences (at the level of donor groups, individual donors, and pancreas regions), and **Fig.3e** provides a succinct summary of distinctive UMAP cluster features. These data are essential to outline the basic outputs of our UMAP clustering strategy and therefore act as a foundation of subsequent analyses in **Figs.4, 5c-e, 6a-c/f/g, 7g-j, S5, S6, S7d/e/h & S8a** as well as our model cartoon in **Fig.8b**. Lastly, while UMAP subclusters are identified and shown in **Fig.3a-c**, they do not constitute a topic of discussion at this stage in the narrative; rather, relevant subcluster properties are shown and discussed in **Figs.4b/d/e & S6c/d**.

In Figure 4, the same data is plotted in A-D just organized differently. We propose in line 315 following that the profound redistribution of relative UMAP cluster magnitudes with disease progression is arguable one of the most relevant observations in our analyses. Accordingly, we devote the entirety of **Fig.4** to this topic. Specifically, **Fig.4a** resolves UMAP cluster display according to pancreas region and T1D stage and thus provides a visual connection to **Fig.3a-c**; ProINS expression is added as a color gradient to facilitate identification of beta cell-containing islets. The heatmap visualization of UMAP cluster magnitudes in **Fig.4b** offers a more granular display at the level of donor groups, individual donors, UMAP sub/clusters and pancreas regions. In short, it is intended to allow for a straightforward appreciation of donor similarities and differences across these parameters. The previous **Fig.4C/D** did indeed “plot the same data just organized differently” for the simple purpose to emphasize statistically significant differences and associated trends. Following the Reviewer’s suggestion, we have removed former panel C from the main figures (now featured in the supplement as **Fig.S6b**), and **Fig.4c** now displays UMAP cluster magnitudes across disease stages.

To better appreciate cluster III subcluster differences, **Fig.4d** provides a succinct overview (that also contains similar information on cluster IV subclusters which are examined in **Fig.S6d**). **Fig.4e**, then, utilizes the same display style as in **Fig.4c** to specifically demonstrate that subcluster dynamics in UMAP cluster III are essentially similar to those in clusters I and II thus emphasizing the shared fates of all ICIs. Lastly, **Fig.4f** translates the relative UMAP cluster dynamics of the preceding panels into their spatial context by calculating respective islet densities. Altogether, our data display seeks to convey the complex UMAP cluster dynamics in a simple fashion,

and the importance of these findings, at least in our view, is emphasized by their contribution to our overall model displayed in **Fig.8b**.

In Figure 5, it's not clear that breaking down the analysis by sub-cluster is of value; it is not truly a finding that immune cells are enriched in specific islet clusters because immune cells were used to define the islet clusters. While subject to investigator opinion, we respectfully disagree with this contention. The Reviewer is correct in that CD45⁺ cell association with islets contributes to cluster segregation; however, it is certainly not the only parameter (details for UMAP clustering are provided in Methods lines 1,092-1,110). The Reviewer's concerns regarding our data on CD45⁺ cell association with islets are addressed in detail in our responses below to items 13.a./c., 14.c./d., 21., 22., Methodology/9. and 10. as well as Other comments (also note that the data in former **Fig.5** is now distributed across the new **Figs.5 & 6** to reduce the overall amount of data panels per figure). Lastly, we do not claim that CD45⁺ cell allocation to specific sub/clusters constitutes a relevant finding in itself; rather, these allocations allow us to focus our downstream analyses in a meaningful fashion as detailed below (cf., **Figs.5d/e, 6c/f/g & S6f-i**).

Figures 6-7 are interesting, but they should be interpreted conservatively. Throughout the manuscript, we aim to advance our data interpretation with due caution. A specific concern pertaining to former **Fig.6** (now **Fig.7**) is articulated by the Reviewer in item 16 where we also provide our reply; however, no specific concern is raised about former **Fig.7** (now **Fig.8**) data interpretation.

6. In the text, I found many of the conclusions very challenging to understand due to the writing. The Reviewer mentions specific instances in the itemized list below, and we provide clarifications and proposed modifications of the phrasing of our conclusions in an effort to improve clarity.

Minor weaknesses

7. Some of the findings have been previously published but are not cited or if cited, the specific findings are not engaged with:

a. Geometric measurements:

i. Seiron et al, measure islet diameter, circularity and islets/mm.
ii. Damond et al, 10.1016/j.cmet.2018.11.014 examine islet solidity and extent over different stages of disease. They also show reductions in both in T1D. Seiron et al. (PMID: 31493350) report unchanged circularity of T1D islets while Damond et al. (PMID: 30713109) find decreased solidity of pseudostage 3 islets. We now provide citations of these reports in line 93 as well as those by Wright et al. (PMID: 32388592) and Tegehall et al. (PMID: 40473678) for other analyses of some shape descriptors in the context of T1D (see item 10. below for further details). The reduction of islet density ("islets/mm²") in T1D reported by Seiron et al. was already discussed in line 212.

b. ProINS/IAPP reduction in AAb

i. Teresa Rodriguez-Calvo et al, <https://doi.org/10.2337/db16-1343> show that proinsulin is increased in AAb+ vs Ctl cases, in contrast to the your data. Potential explanations for this discrepancy should be discussed. We cite Rodriguez-Calvo et al. (PMID: 28137793) in lines 132 and 136 in regard to our distinction of total ProINS and ProINS^{hi} staining areas. Apparent discrepancies pertaining to cumulative ProINS staining areas in the cited reference vs. our work are discussed in lines 769-775, as is a suggested explanation for this matter.

c. Immune infiltration in AAb cases

i. Insulinitis is present in a small fraction of early onset T1D cases and an even smaller fraction of autoantibody-positive cases. In't Veld et al, <https://doi.org/10.2337/db07-0416>. These observations are well established across multiple studies and we cite the review PMID: 34755679 line 349 instead (see also item 7.e. below).

d. Reduction in islet areas with T1D

i. Wilcox et al, 2010, 10.1007/s00125-010-1817-6. Willcox et al. 2010 report mean ICI size in Ctrl donors (~12,000µm²) vs. mean ICI size (~18,000µm²) and IDI size (~9,000µm²) in T1DS donors. However, the

information remains limited since no further breakdown of the data or a discussion thereof are provided. For further details about islet size matters, please see our discussion in items 10., 14.b., 19.a.-c. and 20. below.

e. IDIs with immune cells

i. Barlow et al, ELife, <https://doi.org/10.7554/eLife.100535.3>. An association of immune cells with IDIs has been reported in multiple publications over the past 40 years but quantifications are at times difficult to compare due to varying definitions of insulinitis, selection of islet pools for analysis, and T1D stages interrogated (e.g., Foulis & Stewart 1984 [PMID: 6381192]: ~1% of IDIs are insulitic but insulinitis is not clearly defined; Willcox *et al.* 2009 [PMID: 19128359]: ~5% insulitic IDIs with insulinitis defined as ≥ 5 islet-associated CD45⁺ cells; Damond *et al.* 2019: ~8% of pseudostage 2 IDIs at T1D onset are insulitic using the 2013 consensus insulinitis definition; Barlow *et al.* 2025 [PMID: 36993739]: ~3.8% of “late-stage” IDIs have >2 associated CD8 T cells and >7 macrophages/dendritic cells).

For clarification and comparative purposes, we have now added **Fig.5b**/right panels which display insulinitis frequencies according to ICI and IDI status using the standard insulinitis definition, and we reference review PMID: 34755679 (line 349) in regard to earlier ICI and IDI insulinitis frequency reports. In lines 368-371 we now discuss elevated islet-associated immune cell counts and insulinitis of IDIs with citations of review PMID: 34755679, Damond *et al.* and Barlow *et al.*

f. Clustering of islets regionally

i. Barlow et al, ELife. Unfortunately, we are unsure what is meant with this comment. Barlow *et al.* employ both hierarchical clustering and the dimensionality reduction algorithm UMAP to deconvolute highly-multiplexed immunostaining images, similar to Damond *et al.*, who use both hierarchical clustering and tSNE (Wang *et al.* [PMID: 30713110] also utilize hierarchical clustering for the same purpose). In all publications, analysis modalities pertain to single cells. In contrast, our UMAP analyses are conducted with “single islets”. If the Reviewer is instead referring to spatial “islet aggregations” (the term we previously used to avoid confusion with UMAP “clustering”), this is not something that Barlow *et al.* have addressed (we have now decided to drop the term “aggregation” from the manuscript and to use “density enrichment” instead; however, we use the plural term “aggregates” in lines 430 and 689/**Fig.7b** legend to better explain the concept of “density enrichments”; also *cf.*, “Major claims/item 16.” below).

8. Lack of explanation or even speculation of the biological processes underlying changes in geometric shape descriptors. It would be valuable to show example images of islets and have a pathologist to review them. We now added the speculation in lines 230-231 that alterations of islet architecture are “likely a direct consequence of beta cell destruction”, and we added sample images in **Fig.S1b** featuring four selected islets and their associated shape descriptor values (also, see comment “Methodology/5.” below).

9. I am skeptical that this method will be useful in practice given the significant effort required to perform the staining and image registration. There are now many multiplexed immunofluorescence platforms that seem more feasible. If by “this method” the Reviewer means MICSSS technology, we agree that contemporary immunostaining platforms affording higher analytical dimensionality are likely to become a preferred choice (though residual advantages of MICSSS - such as the use of standard IHC equipment, reagents and protocols; no need for antibody conjugation; generation of pathology-grade bright-field images and/or whole-slide imaging-compatibility - may retain their utility for targeted projects). Importantly, however, we believe that digital pathology and semi-automated analyses of whole-slide pancreas images as well as the deployment of dimensionality reduction algorithms will become ever more important in T1D pathology, and our foray into this field may provide a foundation for future adaptations and extensions. In fact, in lines 538-541/**Fig.S8c** we provide a simple “key” that permits an approximation of islet UMAP clustering as performed here for analysis of archival images of pancreas sections stained only for INS, GCG and CD45 (and PPY for more refined PH analyses); since such analyses can then be conducted with considerably larger donor cohorts, important donor stratifications according to age, age of T1D onset, gender, ethnicity and various clinical variable will be possible to better account for differences in disease course and presentation.

10. It would be very valuable to demonstrate that the geometric changes are not artifacts of the image processing pipeline. This could be done using published highly multiplexed datasets from Damond (Cell Metabolism) et al and Barlow et al (ELife). The generation of islet objects is detailed in Methods lines 1,046-1,068. It employs six hormone stains, yields contiguous objects incorporating non-endocrine areas, and is systematically applied to all islets for the derivation of shape descriptors. It is not clear how the highly multiplexed data sets by Damond *et al.* and/or Barlow *et al.* can be leveraged for the purpose of ruling out artifacts in our image analysis pipeline. For

one, Barlow *et al.* do not report on geometric properties of islets (although they must have determined islet sizes since they state that islet-associated immune cell counts were adjusted for islet size). Damond *et al.* quantified islet solidity across pseudostages and demonstrate a significant decline of this parameter in pseudostage 3 compared to pseudostages 1 and 2. Pseudostages, however, do not strictly correlate with clinical stages and while an approximation (based on additional hierarchical clustering and tSNE analyses of Ctrl and T1D donors) suggests that pseudostage 1 ~ Ctrl, pseudostage 2 ~ T1DS, and pseudostage 3 ~ T1DL (no AAb donors interrogated), the reported solidity values for pseudostage 1 (~0.85), pseudostage 2 (~0.83) and pseudostage 3 (~0.7) (their Fig.3F) are nearly identical to our solidity data for Ctrl and T1DL groups; however, we also find a significant solidity reduction in our T1DS group with a value of ~0.78 (**Figs.2l & S3n**). Lastly, Damond *et al.* also determined islet extent (islet area divided by islet bounding box), a related metric that we have not quantified; we believe that the concurrent decline of islet circularity, solidity and aspect ratio with disease progression as shown in **Figs.2l & S3n** sufficiently describes the progressive degradation of islet shapes. See also above “Minor weaknesses/7.a.” and “Methodology/5.” below.

11. The Discussion section should cite relevant figures with each key claim to make it easier for the reader to evaluate the support for the statement. Figure callouts have been added to the Discussion as requested (though it is not immediately clear if this conforms with *Nature Communications* style policies).

Support for key claims continued

Major claims (broken down point by point from Abstract/Discussion):

12. Line 64-68 are very general and conceptual but not clear. They could be more specific and centered on key findings. We agree that the summary of our findings in lines 64-70 of the Introduction lacks specifics; however, for the purpose of concluding the Introduction section, we have deliberately chosen a more general summary account of our observations since the multiplicity of specific findings reported in our manuscript cannot be convincingly featured without more extensive contextual references. Please note that we have rephrased the respective passage in lines 64-70 to improve clarity.

13. “new histopathological correlates for the stage 1/2 pancreas suggest AAb subjects are “on cusp” of developing T1D hallmarks”: Please see our replies to item 13. subsections below.

a. “higher frequencies of islets with ≥ 1 associated immune cell”. Immune infiltration in AAb cases has been published but it is even rarer than in early-onset T1D so this is hardly a feature of AAb subjects or islets. We certainly agree that islet immune cell infiltration at the AAb stage has been reported previously (and provide citations accordingly), and that it is typically less pronounced than in early-onset T1D. However, we do not claim that immune cell infiltration is a distinctive “feature of AAb subjects or islets”. Rather, we report on a novel metric that robustly captures the increased islet-associated immune cell burden present in the AAb stage, namely the frequency of islets associated with ≥ 1 CD45⁺ cell(s) (**Fig.6e**). A notable advantage of this metric is its simplicity, and it thus can be readily applied to analyses of suitable pancreas section images that cover a sufficient number of islets.

b. “early targeting of small, including “GCG-deficient” islets”. The claim that GCG-deficient islets are targeted early is circumstantial and incomplete (see section 19). Unclear what is meant by “circumstantial and incomplete”. **Fig.2b** demonstrates a relative reduction of all GCG-deficient islets in the PT of AAb, T1DS and T1DL donors that, however, is significant only for T1DS donors (the data display is akin to the typical presentation of IDI frequencies such as shown in the adjacent panel **Fig.2a**). Subsequently, the **Fig.2c** heatmap displays relative magnitudes of four PT islet subsets differentiated according to ProINS/GCG expression across disease course. Here, we find a significant relative magnitude reduction specifically for the ProINS⁺GCG^{neg} subset already in AAb donors, hence our conclusion about the “early targeting of small, including GCG-deficient islets” (while similar patterns are also observed in the PH, significant differences for ProINS/GCG-stratified islet subsets are mostly found in T1DS subjects; **Fig.S3c/d**). Moreover, we have now added a new panel **Fig.2h** that shows a modest reduction of small ICIs lacking alpha, gamma and delta cells at the AAb stage.

c. “collapse of islet cluster I magnitude”. The magnitude of Cluster 1 vs 2 is likely driven by CD45 count so this is the same point as a) above. The Reviewer is correct that one among other features distinguishing UMAP clusters I and II is the presence (cluster II) vs. absence (cluster I) of CD45⁺ cells (*cf.*, **Fig.6a**), and that the readouts in **Fig.6e** (see item a. above) and **Fig.4f** (“collapse of cluster I magnitude”) are related on account of considering islet-associated CD45⁺ cells. The difference, however, lies in the simple enumeration of islets with ≥ 1 associated

CD45⁺ cell(s) (**Fig.6e**; demonstrating a significant difference between Ctrl vs. AAb and T1DS subjects regardless of any UMAP clustering) vs. islet subset stratification according to UMAP clusters (**Fig.4f**; demonstrating a progressive cluster I magnitude reduction across the entire disease course; also note that the UMAP stratification, in addition to cluster I, reveals other clusters without associated CD45⁺ cells [III-A/C, IV-A/C and V-BC]; *cf.*, **Fig.6a**). The latter findings therefore contribute to our revised T1D history model since they suggest an islet “reallocation” from cluster I to cluster II beginning already in the AAb stage (**Fig.8b**).

14. histopathology of T1D:

a. “tissue-wide distributed rather than strictly localized disease process”. It is unclear what supports this. Is this a reference to the similarities across PT and PH? The quoted statement above pertains to the entire pancreas, *i.e.*, PT and PH regions alike. The conclusion draws on multiple strains of evidence shown in the modified **Fig.2n/o/q** as well as **Figs.6f/g, 7, 8a, S4a/c/e/g, S5f/h, S7 & S8**; for details, please see items b.-d. below (also, note that we have added a new **Fig.S5l/J** that summarizes the similarities of UMAP cluster-stratified PT and PH islets, see “Methodology/2.”). Parts of the Discussion have now been rephrased to clarify which data support our contention of a tissue-wide pathogenetic process (lines 513-515 and 521-532).

b. “alterations of islet architectural features in early T1D are independent of islet size” This argues against the claim that that islet size distribution is important. It is not clear what a ‘simple organizing principle’ means practically. The heterogeneity of islets at the level of size, shape, and endocrine composition *etc.* has long been recognized (PMID: 30936150). By considering islet properties as a function of their size, our data in the modified **Figs.2m-q & S4a-h** demonstrate that there is for the most part a simple exponential relation (one-phase exponential association or decay) between islet size and other islet properties (*cf.*, line 236 following; also note that we added a new panel **Fig.2m** showing islet size-dependent INS:GCG expression ratios). We have rephrased the paragraph lines 236-246 in the Results section to improve clarity and restricted usage of the term “organizing principle” to the concluding sentence of that paragraph (lines 245-246). Applying this approach to pathological states permits more granular analyses since a consideration of islet properties in the context of specific islet size bins can reveal disease-associated differences that cannot be discerned or appreciated by a “bulk comparison” of islets. Of the alterations shown in **Fig.2n-q**, the changes in architectural features (islet cell density, solidity and circularity, **Fig.2n/o**), as per the above quoted sentence, are in fact somewhat independent of islet size (*i.e.*, observed for many islet sizes to a similar extent); this finding also contributes to our claim that the disease process is more wide-spread since it affects all islets. In contrast, more pronounced changes can be observed in T1DS for islet hormone composition, and in particular for ProINS, INS and IAPP expression which, in fact, are reduced more obviously in an islet size-dependent fashion (**Fig.2q**). Note that in response to the specific request in “Methodology/6.” below, we made changes to some data analysis and display in **Figs.2n-q & S4a-h** and as a result of these changes, the narrative in lines 236-264/Results and the Discussion has been edited to improve clarity.

c. “absence of a direct correlation between islet composition and associated immune cell abundance”. The fact that small islets without CD45⁺ cells also decrease ProINS expression could be easily explained since only 2D sections are analyzed and immune infiltrate is typically sparse and concentrated at one point. It is reasonable to expect a “dropout effect” for small islets. The Reviewer is correct in raising this interpretive possibility and we state as much in lines 545-547. The fact remains, however, that beta cell content of these islets is reduced (**Fig.6g**), an observation that to our knowledge has not previously been reported, and that constitutes part of our argument that a) it is important to analyze islet properties even in the “2D-absence” of associated CD45⁺ cells as the islets may present with T1D-associated alterations, and that b) the disease process is more wide-spread throughout the pancreas since a relative reduction rather than complete absence of beta cell content in these islets likely has gone unnoticed in prior work and thus contributed to the more focal impression of pathogenic processes.

d. “consider the entire spectrum of immune cell associations with islets beyond those affected by insulinitis”. Unclear. Traditionally, the T1D-specific histopathological hallmark involving immune cells has been insulinitis, the presence of three or more islets associated with ≥15 CD45⁺ cells. Despite its inarguable diagnostic utility, there are obvious limitations to this definition as it features an absolute number of cells in the absence of a specific reference islet size (*cf.*, lines 371-373) and does not consider islets associated with a CD45⁺ cell burden of <15 immune cells. We argue that considering the entire spectrum of immune cell associations ranging from 1 CD45⁺ cell (*e.g.*, **Fig.6e**) to many CD45⁺ cells (*e.g.*, **Figs.6d & 8a**) provides a more inclusive perspective on histopathological alterations that, in present work, provide evidence for our claims about an early diagnostic

metric for AAb donors as well as a more tissue-wide progression of pathogenic disease processes than previously appreciated.

e. Lines 504-508. Unclear. The Reviewer is referring to the following statement: “*And lastly, our analyses of spatial islet distributions foreground the “negative spaces” of T1D histopathology, namely an apparent loss of whole islets from expanding pancreas regions that is accompanied by a contraction of residual areas with “normal” islet density, thus consolidating the concept of a multifocal and more synchronized origin and progression of T1D pathology*”. This sentence from the Discussion pertains to our findings as summarized in the Results section which now has been rephrased for greater clarity in lines 438-441 (further details in item 16. below). If the Reviewer specifically refers to our usage of the term “negative space” in line 525 (in graphic design, the term denotes the “empty space” around an object image; specifically, “negative space” surrounding an object guides visual attention by better revealing the object shape contours), we believe it is an appropriate choice since it readily captures a specific aspect of T1D histopathology: areas in which islets appear to “drop out” (*i.e.*, their local densities are reduced) constitute the “negative space” that grows with disease progression; at the same time, areas with “normal” (*i.e.*, unperturbed) islet densities shrink progressively as they give way to the expanding “negative space”.

Major claims (broken down from Results):

Unclear: please see our clarifications below.

15. Line 178: “absence of alpha cells... suggesting that “GCG-deficient” islets constitute an integral part endocrine pancreas physiology”. If the cited work shows no effect of alpha cells on beta cells, how can they conclude GCG-deficient islets are integral. The cited work PMID: 39169271 demonstrates that human beta cell pseudoislets (*i.e.*, containing no alpha cells) retain the normal glucose-regulated mitochondrial respiration, INS secretion and exendin-4 responses of entire islets. Together with recent 3D mapping of normal pancreata by Lehrstrand *et al.* documenting an abundance of small “GCG-deficient” islets in Ctrl pancreata, the data suggest that “GCG-deficient” islets are not a “histological artifact” but rather, “*constitute an integral part endocrine pancreas physiology*”.

16. The observation in Figure 6 that the distribution of islets can change from being uniform to aggregated is quite interesting but in the exact text in line 428-431 “Since islet locations are immutable... cumulative areas, and mass”, I don’t understand what the specific claim is because the definitions of “tissue regions with unperturbed islet densities” or “islets at shorter distances” is unclear. So, I can’t confidently evaluate the merit of the claim. In the Ctrl pancreas, islets are mostly randomly distributed (*i.e.*, the respective Ripley K curve shows only modest deviations from random distributions corresponding to a value of “1”). With disease progression, some islets “drop out” (*i.e.*, their local density decreases) while adjacent areas with “unperturbed” (*i.e.*, unchanged) islet densities now appear as “aggregates” (although their local islet density has not changed, it is now higher in comparison to adjacent areas with lower densities due to islet drop-out; mathematically, this is reflected in a “density enrichment” in the Ripley’s K function curves shown in **Fig.7b/c** and further confirmed by fractal dimension analysis in **Fig.S7b**). And to be sure, islets of course stay fixed in place, so “density enrichments” emerge by “subtraction” (*i.e.*, loss of islets in neighboring areas) and not by “addition” (*i.e.*, islets moving closer to each other; actually, one might speculate that loss of exocrine tissue in T1D could bring islets into closer proximity with each other; that, however, appears not to be the case since overall Delaunay distances [**Fig.7e**] and Delaunay areas for ICI islet clusters I, II and III [**Fig.7j**] tend to increase and not decrease). In any case, the revised sentence below aims to describe these “density enrichment” dynamics and we further inserted another figure callout to clarify what parts of the sentence refer to which figures (lines 438-444): “*Importantly, since islets do not change their anatomic locations, their density enrichments need to be considered in a relative fashion: because of islet loss at larger distances and a resulting “re-definition” for random distribution at lower islet densities, increasing density enrichments over disease course reflect a progressive reduction of tissue regions with unperturbed islet densities (Fig.7b/c). Altogether, these observations are consistent with the overall decrease of islet densities, cumulative areas, and mass (Figs.1i & S2h) ...*” (also, see item 14.e. above).

Overinterpreted/unsubstantiated: as detailed in our responses below, we believe that our conclusions are readily supported by our data and thus they are neither overinterpreted nor unsubstantiated.

17. line 150 “they emphasize a profound alteration of the endocrine pancreas in the T1DL stage beyond the loss of beta cells”. It’s not clear what is profound about the findings. This conclusion pertains to the data presented in **Figs.1i/j & S2g-k**; we believe that in aggregate, and beyond the loss of beta cells, the reduction of islet densities,

cumulative islet and CHGA areas, islet mass as well as alpha and delta cell mass constitutes a notable alteration of the endocrine pancreas in T1D. Whether this is can be regarded as “profound” or not may remain a matter of debate, so we hope that by using the word “considerable” instead, our assessment will appear less controversial.

18. Line 181: Authors report a trend in ProINS^{hi} islets from Ctrl to AAb (Figure 2D). They would need to show side by side the %ProINS/GCG-negative and %ProINS/GCG-positive islets to say the GCG-deficient are more vulnerable than GCG-positive islets. Former Fig.2D, now Fig.1k (PT), and corresponding Fig.S2I (PH), display a trend toward reduction of the ProINS^{hi} staining area in islets of AAb vs. Ctrl subjects; they do not differentiate between GCG⁺ and GCG^{neg} islet subsets. Line 170 following refers to Figs.2b/c & S3c/d which stratify islet subsets according to ProINS/GCG expression; please see our response to item 13.b. above for details about preferential targeting of small, including GCG-deficient, islets.

19. Line 252: Are Ins- and Ins+ islets included in 2Q (ProINS through SST columns)? Without knowing, it's hard to separate the fraction of beta cells in each islet vs the fraction of islets that are completely insulin deficient. The data in previous Fig.2P/Q, now Fig.2n-q, features islet size-stratifications of combined PT and PH islets within each donor group, i.e., both ICIs and IDIs are included. Together with our earlier demonstration of an apparent loss of small islets (<3,000 μm^2) in the AAb stage (Fig.2g; updated as per item “Methodology/4.” below) and the modest reduction of the ProINS⁺ (i.e., ICI) fraction in that contingent (new Fig.2h), the data in Fig.2q indicate that overall reduced ProINS/IAPP (but not INS) expression likely results from a combination of ProINS/IAPP depletion in particular in the smallest islets and reduced expression of these hormones, especially in slightly bigger islets (3,000-6,000 μm^2 ; also, see adjacent figure where we display relative ProINS content only in PT ICIs). Further analyses of UMAP cluster I ICIs (Figs.3d, 4a & S5b/d) presented later on demonstrate a ProINS/IAPP expression reduction preferentially in small islets even in the histopathological absence of CD45⁺ cells (Fig.6g).

a. “reflects an early loss of beta cells especially in smaller islets”. By “early loss”, do you mean in Ctrl vs AAb or Ctrl vs T1DS? This conclusion follows the callouts to previous Figs.2P/Q & S4C-H where pronounced differences are observed for T1DS vs. Ctrl donors. However, these findings are narratively preceded by our discussion of a small islet loss (islets in the 1,000-3,000 μm^2 size range) already at the AAb stage as detailed in Fig.2g (see update as per item “Methodology/4.” below) and the new panel Fig.2h (also, see above), and we therefore opted in the sentence above for the expression “early loss” since it covers both AAb stage and at-onset T1DS cases.

b. “preservation of residual beta cell mass in larger T1DS islets”. The idea that larger islets preserve beta cell mass is interesting, but the order of events can't be discerned from these data. It is possible that islets shrink after losing beta cells or they all shrink in parallel as disease progresses, not necessarily that smaller islets are targeted first. It would be good to invoke the insulinitis data to make the case that islet size effects targeting. We agree that overall, there is a reduction of median islet size in T1DL vs. Ctrl donors that is significant in the PT but not in the PH (Figs.2j & S3h). However, our temporal reconstruction of pathological events is based on the consideration of islet subsets differentiated according to size and, later on, UMAP cluster allocation (also, see our response to comments 13.b. and 19.b. above where we emphasize the apparent physical loss of islets that are <3,000 μm^2 already in the AAb stage). Please see item 20. below for more details about how the results of our analyses support the above conclusions.

Lastly, it is not clear how invocation of insulinitis data will be helpful at this stage of the narrative. We only address the concept of insulinitis later on (line 342 following) and one of the issues discussed therein is the whiff of circular logic that burdens the insulinitis definition: by setting a threshold in absolute numbers (i.e., ≥ 15 islet-associated immune cells) in the absence of a fixed denominator (i.e., islet size is not specified), the definition privileges larger islets which, due to their size, have a greater probability to be associated with immune cells. Thus, the conclusion that insulinitis is preferentially observed for larger islets, while strictly speaking correct, does not imply that larger islets are, in fact, the preferred autoimmune targets since the insulinitis definition itself is biased towards larger islets (also, cf., Fig.S6h and corresponding legend; regarding insulinitis, see also items 14.d. above and 21. below).

20. Line 304: “early targeting of smaller ICIs and later appearance of larger IDIs” First, it's unclear what is meant by “appearance”. Second “larger” is confusing because IDIs are smaller than ICIs. If I understand correctly, the claim is that the gradual increase in islet size (in ICIs and IDIs) over the course of T1D suggests that smaller

islets are losing insulin and becoming IDIs before larger islets do. Again, I think this is interesting but should not be overstated. If this were the only cause, one would expect IDIs at late stage T1D to be similar size to ICIs (once all islets lose insulin). To be fair, the authors do not claim that size-dependent disease kinetics is the only cause. Conclusions should be qualified and elaborated upon in the discussion. The Reviewer is correct in stating that, despite considerable variability, IDIs tend to be smaller than ICIs; this is implied in the data presentation for PT islets in **Fig.2j** where most islets in Ctrl donors are ICIs and most islets in T1DL donors are IDIs (*cf.*, **Fig.2a**; see also item 19.b. above). However, our islet subset analyses by UMAP clustering complicate this conclusion since, for example, cluster I ICIs have about the same size as cluster V-A IDIs (**Fig.S5a**, left column). At the same time, considering UMAP cluster-specific islet sizes provides analytical opportunities to better understand the “histopathological dynamics” of disease progression.

The sentence quoted above represents our conclusion specifically pertaining to islet property differences within the same UMAP clusters as assessed across disease stages (as opposed to our earlier comparisons of islet properties across different clusters). Specifically, we stated that “... *opposing trajectories for islet size increase vs. circularity decrease in cluster II ICIs and especially cluster IV (PH) and V-A/V-BC IDIs can serve as a histopathological correlate for the dynamics of disease progression (Figs.S5e/g)*”. Focusing, for example, on islet sizes in cluster V-BC (**Figs.S5e/g**, lower left plots; cluster V-BC islets are by definition all IDIs), the data shows that Ctrl islets have a small size whereas AAb, T1DS and T1DL islets demonstrate progressively larger sizes (note that this data display does not account for the numbers of islets in cluster V-BC which are of course very low for Ctrl and very high for T1DL subjects). Now, consider the same data for cluster V-A islets (**Figs.S5e/g**, left, second from bottom; cluster V-A islets are also all IDIs): Ctrl donors do not have any cluster V-A islets; however, they “appear” in the AAb stage and as disease advances, their average sizes become larger. Lastly, note that these “dynamics” are accompanied by a decrease of islet circularity (**Figs.S5e/g**, second column, bottom and second from bottom panels). Based on the implied disease kinetics as illustrated in our model (**Fig.8b**), and assuming that beta cell destruction will never lead to islet size increase (though it may very well may lead to islet size decrease), we interpret the “*opposing trajectories for islet size increase vs. circularity decrease*” with disease progression as an early loss of small ICIs (that become small IDIs after destruction of their beta cells), and a later loss of larger ICIs (that correspondingly become larger IDIs and thereby raise the average size of all IDIs).

We have now rephrased that passage in lines 308-313 to improve clarity. Lastly, please note that this interpretation is not contradicted by our observation in lines 206-207/**Fig.2j** that demonstrate an apparent increase of PT islet sizes in the AAb stage; we explain this “paradoxical” outcome by the physical loss – and not transition to an IDI state - of very small ICIs (**Fig.2g**) such that the median size of remaining islets in fact increases.

21. Line 340-342: 5A looks like there are 3 AAb cases with insulinitis and these are the three that also have CD45+ cells. Text should be clarified to explain that CD45+ cells can be found in specific AAb cases, not that CD45+ cells are elevated at the stage generally. Unfortunately, it is unclear to us what the Reviewer means. We state in lines 344-346 that 3 out of 6 AAb cases present with insulinitis (*i.e.*, having ≥ 3 islets with ≥ 15 associated immune cells) with a callout to **Fig.5a**. We do not claim that “*these are the three [AAb donors] that also have CD45+ cells*” since some immune cell association with islets, albeit to different degrees, is found in all donors. Also, note that we have reallocated data panels from the former **Fig.5** to the new **Figs.5 & 6**; **Fig.5** now features data on insulinitis (including new panels **Fig.5b**/right showing ICI and IDI insulinitis frequencies) and insulinitic islet properties, and **Fig.6** displays the UMAP cluster-stratified and overall islet-associated immune cell burden. Regarding the latter point, we consider the increase of islet-associated CD45+ cell numbers and densities already in the AAb stage noteworthy despite the lack of statistical significance (**Fig.6d**), we discuss the matter in line 390 following, and we describe what we believe to be a rather useful metric in **Fig.6e** that reflects the enhanced islet-associated immune cell burden in stage 1/2 T1D (also see other comments about CD45+ cell analyses above).

22. Line 388: Is there evidence in this dataset that CD45+ cells are recruited to islets directly? For example, are there increases in CD45+ cells in acinar tissue outside islets? If not, it could help substantiate that immune cells are recruited actively at the AAb stage. In the present study, we have only quantified the islet-associated immune cell burden and we document an overall increase, though not significant, of islet-associated CD45+ cell numbers and densities at the AAb stage (**Fig.6d**). However, preceding analyses focused on UMAP cluster II (**Fig.5d**) as well as the development of a novel metric as discussed above (**Fig.6e**) reveal a significant difference already at the AAb stage. An increase of immune cells in exocrine tissue of AAb donors has been previously reported and was found to be significant for CD11c-expressing antigen-presenting cells (Rodriguez-Calvo *et al.* PMID: 24947367); more recent studies quantifying diverse immune cell populations in the exocrine pancreas either did

not include AAb donors (Damond *et al.*) or were limited to only two AAb donors, one of which was a GADA⁺ “stage 0” donor (Barlow *et al.*).

23. Line 443. “Collectively, our observations support the notion that the typically regionalized patterns of T1D represent residuals of a tissue-wide islet depletion, the extent of which is partially obscured by its fundamentally dispersed nature.” Can volumetric data be used to validate this? Is there any evidence in literature or biological mechanism for islets “disappearing” vs growth of exocrine cells? We agree that this is a very interesting question. For example, Seiron *et al.* (PMID: 31493350) report that islet density (*i.e.*, islet numbers per mm²) is significantly reduced in T1DS and T1DL cohorts compared to Ctrl donors. This observation is particularly noteworthy since other studies have observed comparatively reduced pancreas weights/volumes with T1D onset (*e.g.*, PMID: 33471743), and since the vast majority of weight/volume is contributed by exocrine tissue, the “denominator” for islet density calculations is actually smaller in T1D. Seiron *et al.* privilege a data interpretation by which “failure to establish a sufficient islet number to reach beta cell mass needed to cope with increased insulin demand contributes to T1D susceptibility”. Along similar lines, Murrall *et al.* suggest that reduced “endocrine object” densities observed especially in “endotype 1” T1D subjects (more aggressive, early age T1D onset) reflect a potential combination of inherently smaller beta cell mass together with the selective targeting of “small” endocrine objects early in the disease process which results in a subsequent failure to generate “larger” endocrine objects (<https://www.biorxiv.org/content/10.1101/2025.04.11.648319v1>). These studies, just as our own, are limited by their cross-sectional nature, and our own conclusions emphasizing the seeming loss of islets in T1D do not and cannot challenge the possibility that inherently smaller pancreata and associated lower beta cell mass constitute a relevant risk factor for T1D development, and we state as much in lines 545-547.

To our knowledge, volumetric analyses of post-mortem pancreas tissue remain limited to date, but we cite the studies by Lehrstrand *et al.* (only Ctrl pancreata) and Drotar *et al.* (PMID: 38850534) that reports a non-significant trend towards reduced islet densities in pancreatic slices from AAb and T1D donors; a recent preprint by Rippa *et al.* also confirms reduced islet densities and loss of small GCG-deficient islets in 3D analyses of T1DS cases (<https://www.biorxiv.org/content/10.1101/2025.05.14.654045v1>). Definitive resolution of these questions, however, will likely have to await future longitudinal *in vivo* high-resolution imaging studies. Lastly, we have rephrased the above quoted sentence in lines 455-458 to improve clarity.

Methodology:

1. The islet and cell mass calculations are unreliable. IHC is not quantitative enough to evaluate the extremely large dynamic range of hormone secretion. It is better to simply show the pancreatic mass and islet/endocrine cell area separately. Calculation of beta cell mass as done here (multiplication of fractional beta cell area in whole slide images with corresponding pancreas weight) is an admittedly imperfect but nevertheless widely employed practice (*e.g.*, PMID: 26581594 or 22875233; similar calculations are also performed for other endocrine cells and whole islets). We therefore report these data for comparative purposes, and no claims about the dynamic range of hormone secretion are implied. Absolute and relative pancreas weights are featured in **Table S1** and **Fig.S2d**, respectively, and fractional hormone staining areas are shown in **Fig.S2i**.

2. It is unclear the extent to which the differences in percentages between PH and PT throughout the manuscript are due to the PPY islets in the uncinata process. Would it not be clearer to characterize the uncinata process and then exclude this area from all following analyses? We have structured our parallel analyses and discussion of PT and PH regions by first featuring traditional analysis modalities (*e.g.*, quantification of fractional hormone staining areas in tissue sections or islets; islet architecture etc.) to demonstrate, where applicable, the accuracy of our semi-automated image analysis pipeline in relation to the relevant literature (note, however, that quite a few publications interrogating PH sections leave the unique anatomy of the uncinata process unmentioned). Only then do we conduct UMAP and spatial analyses that together reveal that seeming differences between average PT and PH islet properties obtained with traditional experimental readouts are in fact due to admixture of islets with unique properties found in the PH uncinata process. We believe that the overall trajectory of our experimentation and argumentation is sensible, and we have now added a new supplementary Fig.S5i/j that combines a direct comparison of UMAP cluster-stratified PT/PH islet architectural features (taken from the previous Fig.S6A/B) with UMAP cluster-stratified PT/PH islet endocrine properties to provide ready support for our contention of “... an essentially identical endocrine organization across pancreas regions that locates residual differences specifically to the uncinata process in the PH” (Discussion lines 502-504).

3. Panels S3F using Jaccard index (Line 189) are not meaningfully mathematically different from the fraction of those hormones as reported in the main figure. Please note that we undertook the analyses for beta cell

hormones with particular consideration of Leete *et al.* PMID: 32172310 who reported a relative increase of ProINS/INS co-localization (quantified as “Manders overlap coefficient”) in T1DS cases, and we therefore determined the overlap areas of dual combinations of hormones co-expressed by alpha cells (CHGA, ProGCG, GCG) or beta cells (CHGA, ProINS, ProINShi, INS, IAPP) by calculation of Jaccard indices. In the legend to **Fig.S3f**, we note that our observation about reduction of ProINS-INS Jaccard indices with T1D progression appears to contravene the results by Leete *et al.*, but importantly, we provide a succinct discussion invoking both technical considerations as well as study cohort differences to suggest a resolution for this issue.

In regard to the Reviewer’s specific concern about a lack of differences between Jaccard indices and individual hormone staining areas, we agree that for alpha cell hormones (GCG, ProGCG, and CHGA), changes in Jaccard indices (**Fig.S3f**) reflect an increase in relative hormone staining area (cf., **Fig.1k**). However, for beta cells the decrease of the “intersection over union” (IOU) with disease progression is actually greater than the decrease of individual INS, ProINS and IAPP areas. To demonstrate this, we calculated the probable or “expected” IOU for various hormone pairs and then compared it to the measured IOU. The table below shows the ratio of these two values, where low numbers mean the hormones are more segregated than expected by multiplication of the areas (if the Jaccard index were purely a function of the prevalence of each hormone, then the intersection should be the product of the two areas, each normalized to the union area).

		GCG-CHGA	GCG-ProGCG	ProGCG-CHGA	INS-ProINS	INS-IAPP	IAPP-ProINS	INS-CHGA
Ctrl	PH	1.00	0.80	0.92	0.94	0.89	0.98	0.93
Ctrl	PT	1.00	0.81	0.89	0.95	0.84	0.98	0.95
AAb	PH	1.01	0.85	0.95	0.95	0.77	0.99	0.93
AAb	PT	1.00	0.82	0.95	0.94	0.78	0.98	0.96
T1DS	PH	0.90	0.76	0.90	0.65	0.35	0.55	0.90
T1DS	PT	0.98	0.84	0.93	0.74	0.52	0.74	0.94
T1DL	PH	1.09	0.78	0.98				
T1DL	PT	0.98	0.78	0.93				

For alpha cell hormones, the relationship between measured IOU and the calculated “probable” IOU is the same in all T1D states (although GCG-ProGCG was sometimes lower than expected, there is no change with disease progression). In contrast, for beta cell hormones, the measured IOU notably decreases compared to the mathematically “likely” IOU in T1DS cases (and in the AAb case, for the INS-IAPP pair). Notably, there is no decrease in the measured vs. expected ratio for the overlap between INS-CHGA. The text has been updated accordingly in lines 184-189 and 794-801.

4. Figure 2K left vs right. How can AAb have a different islet distribution than the other 3 groups on the left but T1DL has the different distribution than the other 3 in the right? **Fig.2g/left** displays islet frequency distributions, *i.e.*, relative islet numbers (%), whereas **Fig.2g/right** displays islet area distributions, *i.e.*, the product of relative islet numbers (%) and average islet area in respective bins, normalized to 100% for each donor group. While we emphasize a broad similarity of islet frequency and area distribution curves across donor groups, we also note and discuss more subtle disease stage-specific differences. To clarify the latter aspect, we have now adjusted the data display (**Fig.2g** presents data as mean+/-95% confidence interval [CI] error bands, similar to the display in the modified **Fig.2m-q**) and conducted islet size bin-specific ANOVA tests to highlight statistically significant differences. We have modified the narrative in lines 199-205 accordingly to provide a better and more detailed account of our data interpretation.

5. To ensure that differences in sphericity and circularity that accompany T1D are not an artifact of the image processing, (e.g perhaps the reduced hormone expression and imprecise thresholding leads to cell drop out and hence inaccurate measurements of the perimeter and other features). It would be helpful to show multiple images of islets that represent different sphericities, aspect ratios, and solidity. A quantitative validation would also be helpful. For example, since chromogranin is unchanged, you could calculate the geometric shape descriptors only using chromogranin. The shape descriptors as employed here pertain to entire islet structures, not individual hormone staining areas (accordingly, CHGA staining areas cannot be used as islet shape descriptors). The generation of islet objects is detailed in Methods lines 1,046-1,068, employs six hormone stains, yields contiguous objects incorporating non-endocrine areas, and is systematically applied to all islets. As mentioned in “Minor weaknesses/8.”, we have now included the requested sample islet images in **Fig.S1b**.

We regret that it is unclear to us what exactly the Reviewer considers a “quantitative validation”; however, we discuss solidity values as determined in the present study in comparison to those derived by Damond *et al.* in

section “Minor weaknesses/item 10”. Also, note that we do not quantify sphericity; rather, as mentioned in lines 216-219 (now rephrased), we calculate the “spherical equivalent volume” (*cf.*, **Fig.1e**) for all islets captured in the present study, and we feature the summary results in **Fig.S3k** for comparative purposes (and with specific regard to Lehstrand *et al.* PMID: 38632302).

6. 2Q should contain error bars, ideally via bootstrapping. The previous Fig.2P/Q has been reformatted in response to the Reviewer’s request and the data is now presented in Fig.2n-q; similar changes have also been made to the corresponding Fig.S4a-h (that features the same data as Fig.2n-q as well as additional related data in simplified panels to better detail error distributions and exponential curve fits). In brief, the previous panels plotted donor group means of islet properties from combined PT/PH islets against islet size bins. We have now calculated donor-specific islet property means allowing for display of errors (95% CI shown as error bands) throughout Figs.2m-q & S4a-h, and we performed islet size bin-specific ANOVA tests as detailed in the modified figure legend to Fig.2n-q (note that we have removed the T1DL GCG and SST data from respective Fig.2p panels to avoid crowding; those data can be found in Fig.S4h). As a result of these changes, the narrative in lines 235-264 has been edited accordingly.

7. In Figure 2D, is the z-score computed column wise or row wise, and separately or pooled between PT and PH islets. It appears that the Reviewer is referring to the heatmap in **Fig.2c** rather than the scatter plots in the previous **Fig.2D**. The heat map does not display z-scores but rather the relative magnitude (mean percentages) of indicated ProINS/GCG islet subsets in respective donor groups (*i.e.*, it is organized “row-wise”); the data pertains to PT islets only (we state in the figure legend that all summary plots pertain to the PT, unless mentioned otherwise); and corresponding PH data are found in **Fig.S3d** (similar considerations apply to the heatmap in **Fig.2e**).

8. Is figure 4A intended to show the frequency of islets in each cluster or the ProINS expression over time. It doesn’t serve either purpose well in its current form. We respectfully disagree with this assessment. Fig.4a, in contrast to Fig.3b/c (which features all islets combined from all donors and pancreas regions), stratifies the UMAP display of islets across pancreas regions (PT and PH) as well as disease stage (Ctrl, AAb, T1DS, T1DL); the additional color-coding of ProINS expression serves to highlight the presence of ICIs. As such the figure is intended to demonstrate the implied dynamics of islet UMAP cluster allocation with disease progression, and we believe that the chosen visual display achieves this purpose (see also “Major weaknesses/item 5.” above).

9. Are only hormones used in islet UMAP or geometric features as well? UMAP analyses, including a summary of all input features utilized in the present analyses, are detailed in the Methods section under the header “CytoMAP analysis” lines 1,092-1,110.

10. In Figures 3-4, It’s essential to show the features that were used for the umap and their means in each cluster. The umap uses the islet area and CD45+ cell count which clearly will separate clusters 1 and clusters 2 but this was not clear at all from the text or figures. As mentioned in item 9. above, UMAP analysis details are provided in the Methods section lines 1,092-1,110. The heatmap-adjacent histograms in Figs.3d & S4j display islet feature distributions combined from all donors islets stratified according to cluster allocation (*i.e.*, corresponding to the UMAPs in Fig.3c). We believe that presenting feature distributions here, rather than means, is more informative. The heatmaps, in turn, display donor-stratified means of indicated islet properties for each major cluster I – V-BC. Fig.3e provides a succinct summary of distinctive architectural and endocrine UMAP cluster features, including the different sizes, circularities and ProGCG/GCG content between cluster I and II islets. Lastly, the Reviewer is correct that another feature distinguishing clusters I and II is the absence (cluster I) or presence (cluster II) of CD45+ cells; these data are shown in Fig.6a and they are discussed in depth in line 373 following and line 402 following (see also rest of Figs.5 & 6).

11. Figure 5E needs statistics. Statistical differences between properties of insulitic vs. non-insulitic islets were calculated with paired Student’s t-test using data from individual AAb and T1DS PT and PH tissue sections as detailed in the modified figure legend, with asterisks added where applicable to the histogram panels in Fig.5e. Also, we have rephrased the narrative in lines 356-361 to clarify that insulitic islets, other than being larger and exhibiting lesser cellular density, are not fundamentally different from non-insulitic islets.

12. In Figure 7, the use of Delaunay triangulation and ‘isolated islets’ does not effectively describe the regionalization of islet clusters because it is highly influenced by the abundance of the clusters. A manual approach could include repeatedly sampling neighboring islets vs random islets from the same tissue and compare the enrichment of a given cluster in the sample. Alternatively, a spatial correlation method such as a

cross-K function would be more direct and appropriate. We agree that Delaunay triangulation alone is not sufficient to fully describe spatial islet organization (please note that Delaunay triangulation is discussed in former Fig.6 rather than former Fig.7, and which are now represented as Figs.7 & 8). We therefore conducted density enrichment calculations based on Ripley's K function as shown in Figs.7b/c & S7a. Here, the distances between islets are normalized to the total islet density in each section individually; accordingly, the enrichment shows how much higher the density is around the average islet compared to the whole tissue density. We strongly believe that automation of this task is preferable to manual analyses since it compares every islet within the tissue section with every other islet therein, something that cannot be recapitulated manually.

Other comments:

Is any prescreening performed on the tissues for insulinitis before selecting blocks for the study? If so, this minimizes the ability to use CD45+ cells as a diagnostic as proposed in line 387. As mentioned in lines 49-51, pancreata from stage 1/2 donors and those at onset of stage 3 are exceptionally rare. Given their importance for natural history T1D studies, we therefore worked with the nPOD organ procurement core to provide us with as many of those at-risk and at-onset donor samples as possible; we also included donors with very short duration of clinical disease (both at onset and short-duration T1D cases constitute our T1DS group), and Ctrl as well as T1DL donors were matched accordingly as summarized in lines 992-995. Hence, while we did not engage in pre-screening, AAb donor 6310, despite having only a single autoantibody and therefore classified as "stage 0", was specifically added due to the presence of "low-grade" insulinitis as determined by nPOD pathologists. However, our semi-automated image analyses of PT and PH tissue sections made available to us did not confirm this diagnosis. All of this information is provided in **Tables 1 & S1**. Lastly, deliberate exclusion of donor AAb 6310 from ANOVA tests in Fig.6e does not abolish significant differences between Ctrl, AAb or T1DS groups as shown in the adjacent figure.

Do islet cluster foci in Figure 7 comport to lobules boundaries? This is an interesting question (the Reviewer is likely asking specifically about "insulitic islet" foci in Fig.8a), and visual inspection appears to suggest that these foci do not necessarily comport to lobule boundaries. However, we have not yet been able to incorporate a robust demarcation of lobule boundaries into our analysis pipeline which is necessary to assess this hypothesis in an unbiased and comprehensive fashion.

To validate the utility of geometric shape descriptors, can they be measured in HandE images and applied to large datasets? Yes, this could be done and we advocate for future application of our image analysis pipeline, some modifications notwithstanding, to analyses of archival images in lines 538-541/Fig.8c. We would like to emphasize that the segmentation workflow developed here for MICSSS images does not directly apply to H&E images, and developing a new segmentation algorithm of H&E-stained pancreas images is beyond the scope of this work. As mentioned above, however, future studies can certainly leverage our object analysis pipeline to measure shape descriptors in other datasets.

Reviewer #4 (Remarks to the Author):

I co-reviewed this manuscript with one of the reviewers who provided the listed reports. This is part of the Nature Communications initiative to facilitate training in peer review and to provide appropriate recognition for Early Career Researchers who co-review manuscripts. We thank the Reviewer for their efforts.

Reviewer #4 (Remarks on code availability):

I am not able to assess the code, as it falls outside my technical expertise and background. Therefore, I cannot evaluate the reproducibility of the results or the usability of the code for the broader community.

Response to Referees

Once again, we wish to thank the Reviewers for their time and effort to assess our manuscript, and we in particular appreciate the additional task taken on by Reviewers #1 and #2 to evaluate the extensive discussion of Reviewer #3 comments. Below, please find our point-by-point reply in which the callout to manuscript lines refers to the newly revised manuscript (Reviewer comments are in black font, our replies are in blue font). In the revised manuscript itself, changes are highlighted by gray background shading to readily identify the edits made throughout the entire text.

Reviewer #1 (Remarks to the Author):

The Authors have satisfactorily addressed all the points raised by this reviewer. They are nevertheless invited to succinctly mention the issues discussed in their responses to point 4 (co-stainings) and point 5 (beta cell dedifferentiation) in the manuscript, for the sake of completeness. We thank the Reviewer for this assessment, and we now emphasize the limitations of our experimental approach in lines 662-668 by detailing why MICSSS technology is not the method of choice for robust identification of bi- or tri-hormonal endocrine cells in islets, and why we cannot make any statements about potential beta cell dedifferentiation.

Reviewer #1 (Remarks on code availability):

The code can be now better appreciated. We thank the Reviewer for this assessment.

Reviewer #2 (Remarks to the Author):

The manuscript has been revised. Figures have been modified in line with reviewers' recommendations, and a supplementary table has been included to present the quantitative data underlying Fig. S2e, Fig. S2g, and Figs. 1i and S2h (including total tissue area, parenchymal tissue area, and the number of islets analyzed per section). The issues raised in comments 3 and 4 have been comprehensively addressed, and the response to comment 2 is convincing. We thank the Reviewer for this assessment.

Reviewer #3 comments mediated by reviewer #1 and #2

I have gone through the comments by Reviewer #3 and the responses by the the Authors. The comments are very detailed and (generally speaking), pertinent. In some cases they are not fully clear, and in a few instances look somewhat repetitive. Having said that, the Authors have seriously taken into consideration the many points raised, and to my opinion they have provided reasonable and satisfactory answers.

All the comments raised by Reviewer #3 have been addressed, either through amendments to the manuscript or by providing clarifications in the rebuttal. We thank the Reviewers for these assessments.

Reviewer #4 (Remarks to the Author):

I co-reviewed this manuscript with one of the reviewers who provided the listed reports. This is part of the Nature Communications initiative to facilitate training in peer review and to provide appropriate recognition for Early Career Researchers who co-review manuscripts. We thank the Reviewer for their effort.